# Genomic surveillance of antimicrobial resistance shows cattle and poultry are a moderate source of multi-drug resistant non-typhoidal *Salmonella* in Mexico

Enrique Jesús Delgado-Suárez[1], Tania Palós-Guitérrez[1], Francisco Alejandro Ruíz-López[1], Cindy Fabiola Hernández Pérez[2], Nayarit Emérita Ballesteros-Nova[1], Orbelín Soberanis-Ramos[1], Rubén Danilo Méndez-Medina[1], Marc W. Allard[3], María Salud Rubio-Lozano[1]*

1 Facultad de Medicina Veterinaria y Zootecnia, Universidad Nacional Autónoma de México, Ciudad de México, México, 2 Centro Nacional de Referencia de Plaguicidas y Contaminantes, Dirección General de Inocuidad Agroalimentaria, Acuícola y Pesquera, Servicio Nacional de Sanidad, Inocuidad y Calidad Agroalimentaria, Estado de México, México, 3 Division of Microbiology, Office of Regulatory Science, Center for Food Safety and Applied Nutrition, U. S. Food and Drug Administration, College Park, Maryland, United States of America

* msalud65@gmail.com

**Data Availability Statement:** All relevant data are within the paper and its Supporting information files. Raw sequences are publicly available at NCBI

## Abstract

Multi-drug resistant (MDR) non-typhoidal *Salmonella* (NTS) is a public health concern globally. This study reports the phenotypic and genotypic antimicrobial resistance (AMR) profiles of NTS isolates from bovine lymph nodes (n = 48) and ground beef (n = 29). Furthermore, we compared genotypic AMR data of our isolates with those of publicly available NTS genomes from Mexico (n = 2400). The probability of finding MDR isolates was higher in ground beef than in lymph nodes:$\chi^2$ = 12.0, P = 0.0005. The most common resistant phenotypes involved tetracycline (40.3%), carbenicillin (26.0%), amoxicillin-clavulanic acid (20.8%), chloramphenicol (19.5%) and trimethoprim-sulfamethoxazole (16.9%), while more than 55% of the isolates showed decreased susceptibility to ciprofloxacin and 26% were MDR. Conversely, resistance to cephalosporins and carbapenems was infrequent (0–9%). MDR phenotypes were strongly associated with NTS serovar ($\chi^2$ = 24.5, P<0.0001), with Typhimurium accounting for 40% of MDR strains. Most of these (9/10), carried *Salmonella* genomic island 1, which harbors a class-1 integron with multiple AMR genes (*aadA2*, *bla-CARB-2*, *floR*, *sul1*, *tetG*) that confer a penta-resistant phenotype. MDR phenotypes were also associated with mutations in the *ramR* gene ($\chi^2$ = 17.7, P<0.0001). Among public NTS isolates from Mexico, those from cattle and poultry had the highest proportion of MDR genotypes. Our results suggest that attaining significant improvements in AMR meat safety requires the identification and removal (or treatment) of product harboring MDR NTS, instead of screening for *Salmonella* spp. or for isolates showing resistance to individual antibiotics. In that sense, massive integration of whole genome sequencing (WGS) technologies in AMR surveillance provides the shortest path to accomplish these goals.

and the accession numbers and metadata are available in supplementary S1 File. We have also included the doi number for the two laboratory procedures that are citable and have been uploaded at protocols.io.

**Funding:** This research was funded by the National Autonomous University of Mexico (www.unam.mx), grant number PAPIIT IN212817, awarded to MSRL and OSR. Whole genome sequencing of isolates was conducted free of charge by the Mexican Department of Agriculture (Secretaría de Agricultura y Desarrollo Rural, Centro Nacional de Referencia de Plaguicidas y Contaminantes, Dirección General de Inocuidad Agroalimentaria, Acuícola y Pesquera, Servicio Nacional de Sanidad, Inocuidad y Calidad Agroalimentaria) (https://www.gob.mx/senasica/acciones-y-programas/centro-nacional-de-referencia-de-plaguicidas-ycontaminantes- cnrpyc). The funders had no role in study design, data collection and analysis, decision to publish, or preparation of the manuscript. There was no additional external funding received for this study.

**Competing interests:** The authors have declared that no competing interests exist.

## Introduction

For decades, experimental data have supported concerns that antimicrobial use (AMU) in animal production is a key factor contributing to the ever-growing crisis of bacterial AMR. However, there is limited evidence linking food consumption with AMR emergence in humans [1, 2]. Moreover, an increasing number of studies in developed countries have found that AMU in food animals has a limited impact on the AMR profile of foodborne pathogens [3–6]. However, this situation is probably different in low and middle-income countries (LMIC), where controls of AMU in veterinary practice and human health are less rigorous. Therefore, in the context of an intense trade of foods between countries differing in AMR food safety, it is vital to identify relevant sources of AMR pathogens along the food chain. This measure would help prevent their dissemination, as well as human exposure to MDR bacteria, which is a global public health issue.

In Mexico, source attribution of foodborne illnesses is at an early stage. Nonetheless, there are reports of high rates of infections commonly transmitted through foods. That of non-typhoidal salmonellosis has been above 60 cases per 100 thousand inhabitants in the last 5 years [7]. In addition, recent studies conducted in Mexico show NTS contamination is unusually high (15 to nearly 100% positive samples) in bovine lymph nodes, beef carcasses and ground beef, with isolates exhibiting rates of resistance that vary from 30 to nearly 98%, across several important antimicrobial classes (i. e. penicillins, aminoglycosides, cephalosporins, quinolones) [8, 9]. This evidence suggests that beef is likely to play a role as a relevant reservoir of foodborne MDR salmonellosis in Mexico. Especially, if considering it is often involved in salmonellosis outbreaks in countries with lower NTS contamination rates [10, 11].

The increasing availability of whole genome sequencing (WGS) technologies has helped improve genomic surveillance as it provides a high-resolution method for the characterization of an organism features. Regarding AMR in NTS, however, most research from LMIC deals with phenotypes [12–16]. Although these studies raise concerns, they do not provide insights into the genetic basis of AMR, its evolution or dissemination within bacterial populations. Such information is crucial to devise new strategies to contain the dissemination of AMR pathogens. However, it can best be obtained by addressing phenotypic and genomic profiling of AMR simultaneously, an area with a limited number of studies in Mexico and other countries [17].

In this investigation, we conducted antibiotic susceptibility testing and WGS of 77 NTS isolates collected during a previous research project involving bovine lymph nodes (n = 800) and ground beef (n = 745) across a two-year sampling period [18]. Assembled genomes were used to predict the AMR genomic profiling of NTS isolates, and these data were further compared to their corresponding AMR phenotypes. We also conducted comparative genomics of AMR genotypes of publicly available NTS strains isolated in Mexico from human, bovine, avian, vegetables and produce, water, seafood, and other sources. Consequently, we managed to thoroughly characterize the role of cattle as a reservoir of NTS harboring AMR genes of human clinical significance. Moreover, we identified the dominant genetic determinants that sustain resistance to specific antibiotic classes, as well as MDR phenotypes.

## Materials and methods

Animal Care and Use Committee approval was not obtained for this study since live animals were not directly involved in the experiment.

### *Salmonella* isolates

Isolates used in this research (n = 77) originated from a previous study conducted by our research group [18]. In that work, isolates were subjected to WGS, *in silico* serovar prediction

and multi-locus sequence typing (MLST). To describe important aspects of these isolates, we are going to provide some details of the above referred publication [18].

The isolates were obtained from bovine lymph nodes (LN, n = 800) and ground beef (GB, n = 745), collected from carcasses at a wholesale store in Mexico City across a two-year period (April 2017 through December 2018). The carcasses sold in this store come from a vertically integrated company (feedlot and slaughter operations) located in the state of Veracruz, Mexico. The slaughter population is composed of crossbred *Bos indicus* young bulls (24–36 months of age). Peripheral LN (superficial cervical and subiliac), deep LN (axillary and celiac), lean meat and fat trimmings were collected from each carcass and subjected to *Salmonella* spp. detection and isolation procedures. A full description of *Salmonella* analyses is available from protocols.io (dx.doi.org/10.17504/protocols.io.bpybmpsn). Each isolate was obtained from a different sample across the two-year sampling period.

Pure *Salmonella* isolates were subjected to WGS. Briefly, we extracted genomic DNA (gDNA) from fresh colonies grown overnight at 37˚ C in tryptic soy broth. For that purpose, we used the Roche PCR High Purity Template Preparation Kit (Roche México, Mexico City, Mexico), according to manufacturer's instructions. Subsequently, gDNA was quantified using a Qubit 3.0 Fluorometer (Thermo Fisher Scientific México, Mexico City, Mexico). Next, sequencing libraries were prepared from 1 ng gDNA using the Nextera XT Library Preparation Kit v.3 (Illumina) and sequenced on the Illumina NextSeq system (paired end 2 x 150 bp insert size). Raw sequences are publicly available at the National Center for Biotechnology Information (NCBI). The accession numbers and metadata are listed in S1 File.

The obtained raw reads were then used for *in silico* serovar prediction (SeqSero software, version 1.2) [19] and multi-locus sequence typing (MLST) [20]. Both analyses were conducted at the Center for Genomic Epidemiology website (http://www.genomicepidemiology.org). We identified eight *Salmonella* serovars: Anatum (n = 23), Reading (n = 23), Fresno (n = 4), Typhimurium (n = 11), London (n = 9), Kentucky (n = 6), and Muenster and Give (one each). Moreover, MLST showed isolates of the same serovar corresponded to the same sequence type (ST), except Typhimurium. Among the identified STs, those of serovar Kentucky (ST-198) and Typhimurium (ST-19 and ST-34) are epidemiologically relevant and often exhibit MDR phenotypes [21–23].

During the present research, after assembling genomes, we discarded one serovar Reading isolate that had a poor assembly quality and yielded inconsistent results with *Salmonella* species (genome size >8 Mb and GC content of 46.5%). Consequently, this isolate was not uploaded to NCBI. We also repeated serovar prediction with assembled genomes using SeqSero2 [24] and SISTR [25]. Results were mostly the same, except for one serovar Typhimurium isolate that was actually a monophasic variant (1,4,[5],12:i:-). Hence, this study reports the correct serovar for this isolate (NCBI biosample accession SAMN12857424). Notice this isolate may still be recorded as serovar Typhimurium at NCBI, as it was submitted before we made the correction. However, we expect this record to be updated soon at the NCBI pathogen detection website.

Overall, we managed to set up a panel of strains collected from epidemiologically related bovine matrices (lymph nodes and ground beef), across two years and representing different *Salmonella* serovars and STs. Thus, providing the basis for a thorough characterization of the AMR profiles of non-clinical isolates circulating in beef cattle.

## Antibiotic Susceptibility Testing (AST)

The phenotypic AMR profile of NTS isolates was determined by a panel of 14 antibiotics included in the World Health Organization (WHO) list of critically important and highly

important antimicrobials [26]. Following WHO's prioritization criteria, we selected antibiotics that are considered critically important (i. e. aminoglycosides, quinolones, penicillins, third and fourth generation cephalosporins, carbapenems) or highly important (i. e. phenicols, folate pathway inhibitors, tetracyclines). Moreover, we prioritized antimicrobials that are currently approved in Mexico for use in both human and veterinary medicine. Although azithromycin is frequently used to treat *Salmonella* infections, we excluded macrolides in the AST panel. This decision was made for three reasons:

1. macrolide resistance usually involves typhoidal strains [27],

2. macrolides are not approved for use in food-producing animals in Mexico [28], and

3. azithromycin-resistant *Salmonella* has not been isolated from foods (including meats) in Mexico in the last two decades [9].

Hence, macrolides were considered only for comparative genomics of AMR profiles. For the AST analysis, we used the disk diffusion method [29] with the Bencton Dickinson disks and concentrations reported in Table 1. Isolates were classified as susceptible, intermediate or resistant, according to the Clinical Laboratory Standards Institute (CLSI) guidelines [30]. Strains of *Escherichia coli* ATCC 8739, *Enterococcus fecalis* ATCC 29212, *Staphylococcus aureus* ATCC 25923, and *Pseudomonas aeruginosa* ATCC 9027 were used as quality control organisms. Isolates with intermediate and resistance phenotypes were considered as non-susceptible

**Table 1. Antimicrobial agents tested, their concentrations and clinical break points of the zone diameter used in the AST.**

| Antimicrobials | C, µg[2] | Zone diameter breakpoint[1], mm | |
|---|---|---|---|
| | | R | I |
| Aminoglycosides | | | |
| Amikacin (AMK) | 30 | $\leq 14$ | 15–16 |
| Gentamicin (GEN) | 10 | $\leq 12$ | 13–14 |
| Penicillins | | | |
| Carbenicillin (CB) | 100 | $\leq 19$ | 20–22 |
| Amoxicillin/clavulanic acid (AMC) | 20/10 | $\leq 13$ | 14–17 |
| Third generation cephalosporins (3GC) | | | |
| Cefotaxime (CTX) | 30 | $\leq 22$ | 23–25 |
| Ceftriaxone (CRO) | 30 | $\leq 19$ | 20–22 |
| Fourth generation cephalosporins (4GC) | | | |
| Cefepime (FEP) | 30 | $\leq 18$ | 19–24 |
| Carbapenems | | | |
| Imipenem (IPM) | 10 | $\leq 19$ | 20–22 |
| Ertapenem (ETP) | 10 | $\leq 18$ | 19–21 |
| Meropenem (MEM) | 15 | $\leq 19$ | 20–22 |
| Chloramphenicol (CHL) | 30 | $\leq 12$ | 13–17 |
| Trimethoprim-sulfamethoxazole (SXT) | 1.25/23.75 | $\leq 10$ | 11–15 |
| Ciprofloxacin (CIP) | 5 | $\leq 20$ | 21–30 |
| Tetracycline (TET) | 30 | $\leq 11$ | 12–14 |

[1]Criteria to consider isolates as clinically resistant (R) or intermediate (I) [30]. For meropenem, we also considered epidemiological cutoff (ECOFF) values set by the European Committee on Antimicobial Susceptibility Testing (EUCAST) to classify strains as wild-type (>27 mm) or non-wild-type ($\leq$27 mm) [32].

[2]Antimicrobial's disk concentration.

in this study. Likewise, isolates showing resistance to ≥3 antimicrobial classes were classified as MDR [31]. The detailed AST protocol is available at protocols.io (dx.doi.org/10.17504/protocols.io.bpypmpvn).

## Genome assembly

The quality of raw reads was first assessed with FastQC [33] and we used Trimmomatic [34] to filter poor-quality reads and Illumina adaptors. Trimmed sequences were analyzed again with FastQC to ensure that only high-quality reads (i. e. Q≥30) were used for *de novo* genome assembly. Finally, we used the Pathogen Resource and Integration Center (PATRIC) web server [35] to assemble genomes with SPAdes version 3.13.1 [36]. The quality of genome assembly was assessed within PATRIC with the aid of the QUAST program, version 5.02. Data on genome assembly quality are provided in S1 File.

## Genotypic AMR profiling and comparative genomics

AMR genes and point mutations associated with AMR were predicted with the aid of AMR-FinderPlus program version 3.8.4 using assembled genomes [37]. The study also compared the genetic AMR profile of our isolates in the context of NTS population circulating in the country. For that purpose, we identified all *Salmonella* isolates from Mexico that were publicly available at NCBI as of September 21, 2020 (n = 2400). For that purpose, we worked at the NCBI Pathogen Detection website (https://www.ncbi.nlm.nih.gov/pathogens) and used "organism" (*Salmonella enterica*) and "location" (Mexico) as the filtering criteria. After removing typhoidal strains, we conformed groups of isolates sharing a common isolation source: human (n = 32), bovine (n = 179, including 77 from this study), avian (n = 193), seafood (n = 131), papaya (n = 279), pepper (n = 216), other vegetables (n = 568), surface water (rivers, ponds, lakes, dams, n = 307), and other water sources (n = 246). An additional category named "other sources" (n = 249) included isolates from sources with few records (i. e. animal feed, pet food, dietary supplements, goat, etc.), as well as those with ambiguous (i. e. product, sponge, swab, meat, animal feces, etc.) or unreported isolation source. The full list of accessions and metadata of these isolates is provided in the S2 File. AMR genotypes of these isolates were collected from the NCBI Pathogen Detection website, which are generated with the AMRFinder-Plus database and program. Some of these data were obtained with different versions of AMRFinderPlus (i. e. 3.2.3, 3.6.7, 3.8.4, and 3.8.28). Hence, we repeated the analysis for these records with version 3.8.4, to make sure there was no impact on the AMR genotype prediction.

Most serovar Typhimurium isolates (8/10) exhibited a penta-resistant phenotype similar to that reported for the Typhimurium DT104 strain, which is sustained by *Salmonella* genomic island 1 (SGI1) [23]. Hence, we conducted a Basic Local Alignment Search Tool (BLAST) atlas analysis to determine whether the MDR profile of these isolates was associated with this genomic feature. For that purpose, we used the assembled genomes at the GView web server [38], configured as follows: expect e-value cutoff = 0.001, genetic code = bacterial and plant plastid, alignment length cutoff = 50, percent identity cutoff = 70 and tblastx as the BLAST program. The SGI1 reference sequence (AF261825.2) was collected from the Pathogenicity Island Database [39]. Furthermore, considering the epidemiological importance of this serovar, we also analyzed the whole set of Typhimurium isolates from Mexico deposited at NCBI. For that purpose, we used organism (*Salmonella enterica*), serovar (Typhimurium), and location (Mexico) as filtering criteria. In this way, we identified 40 Typhimurium isolates in the database (refer to S3 File for their accession numbers and AMR genotypes). This analysis was performed to estimate how common MDR profiles are in strains of this serovar circulating in Mexico.

## Plasmid profiling

Plasmids are known as strong contributors to AMR dissemination. Since draft genomes contain both chromosomal and plasmid DNA, we decided to conduct plasmid profiling of our isolates. To achieve this goal, we used an *in silico* approach that has been previously described [40, 41]. First, we used PlasmidFinder 2.1 [42] to predict the isolates plasmid profile using assembled genomes and a threshold identity of 95%. In case of positive hits, we downloaded the plasmid reference sequence from NCBI and aligned it to the draft genomes. If most of the plasmid sequence was represented in a genome and the genomic context of genes matched that of the plasmid, these isolates were proposed to carry the predicted plasmid. Finally, if contigs did not tile well against reference plasmids, we also looked at the average depth of coverage to infer the presence of plasmids in that genome. Usually, the read depth of plasmid-associated contigs is similar between them and different from that of chromosomal contigs.

## Data analysis

We used Chi-square tests and odds ratio (OR) calculations to determine whether there was an association between AMR profiles of isolates and NTS serovar or isolation source. In experimental isolates, these analyses involved AMR phenotypes and genotypes. We also calculated the Pearson correlation coefficient between the number of phenotypic non-susceptible isolates and the number of genotypic non-susceptible isolates for each tested antibiotic. In public isolates, only AMR genotypes were considered, and we analyzed these data with the aid of a heatmap. For that purpose, we first determined the number of isolates from each source harboring specific AMR genes and having MDR genotypes. Using these figures, we calculated the proportion of isolates from each source having the feature and used these data to generate a heatmap with the Heatmapper software [43].

# Results

## Phenotypic and genotypic antimicrobial resistance profiles

Approximately three-quarters of the isolates were non-susceptible to at least one antibiotic class, while 26% were susceptible to all the tested antibiotics (Fig 1). The most common resistant phenotypes included tetracycline (40.3%), carbenicillin (26.0%), amoxicillin-clavulanic acid (20.8%), chloramphenicol (19.5%) and trimethoprim-sulfamethoxazole (16.9%). Resistance to cephalosporins and aminoglycosides was less frequent (1.3–7.8% across these antibiotic classes), while for carbapenems only intermediate resistance was observed at a frequency of 1.3 and 9.1% for ertapenem and imipenem, respectively. Although only one isolate resisted ciprofloxacin, 54.5% of the strains showed intermediate resistance to this drug.

The rate of MDR strains was 26%, with the most common MDR profiles involving penicillins, chloramphenicol, trimethoprim-sulfamethoxazole, ciprofloxacin and tetracycline. Strikingly, one serovar Typhimurium isolate resisted all antibiotic classes but carbapenems. In fact, the probability of isolates with MDR profiles was significantly higher in strains of serovar Typhimurium compared to other serovars (OR = 45.8, 95% confidence interval [95CI] 5.3–399.2, P<0.0001). Likewise, there was a higher probability of finding MDR strains in ground beef compared to lymph nodes (OR = 6.5, 95CI 2.1–20.1, P = 0.0005). Furthermore, isolates collected in 2018 were more likely to have MDR phenotypes compared to those from 2017 (OR = 3.4, 95CI 1.2–10.0, P = 0.02). Similarly, isolates collected in autumn were more likely to have MDR phenotypes compared to those from other seasons (OR = 5.8, 95CI 1.8–19.1, P = 0.0054). Overall, the WGS-based *in-silico* prediction of non-susceptible phenotypes was good, as shown by the strong association between AMR genotypes and phenotypes (Fig 2).

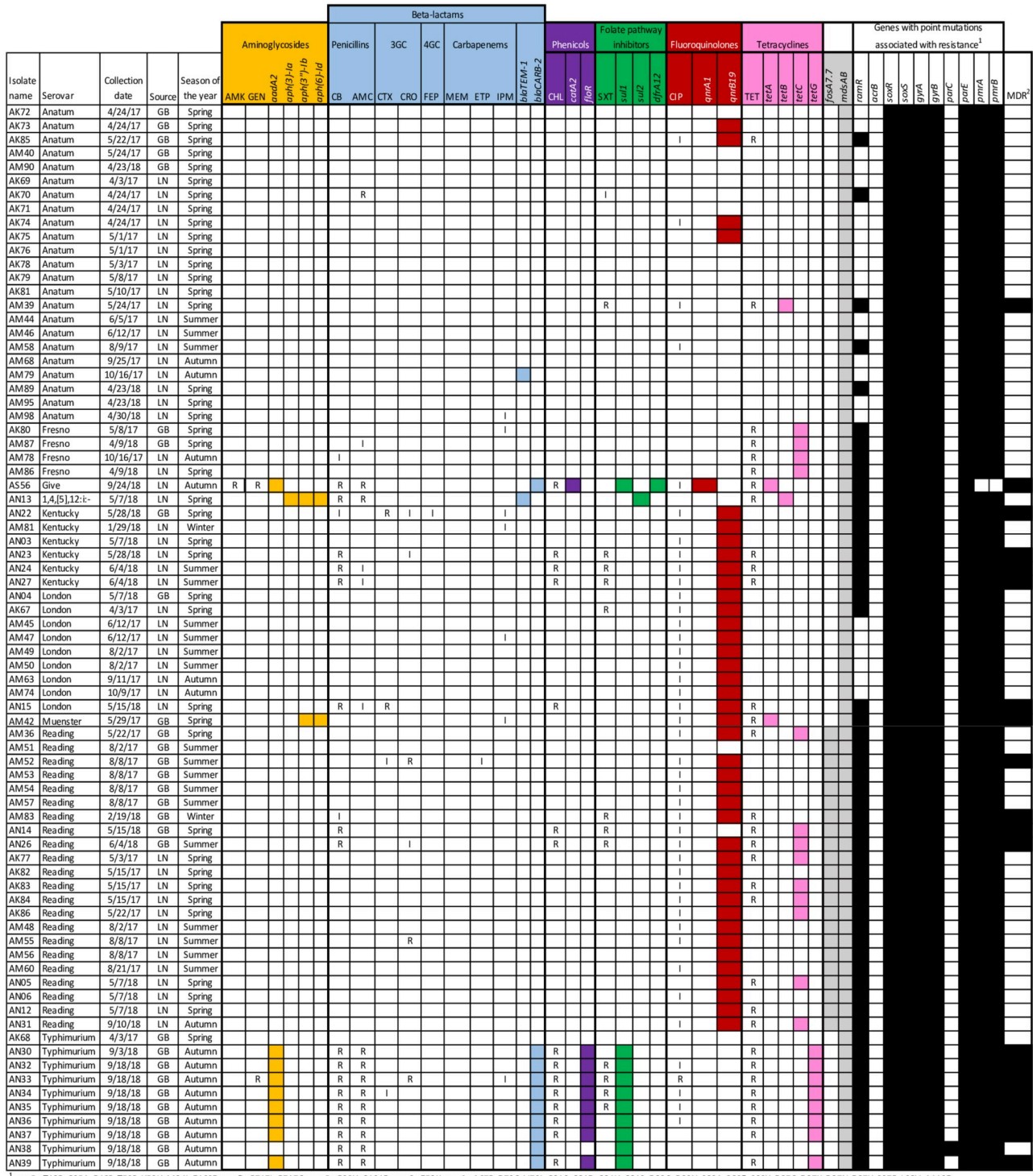

**Fig 1. AMR profile of 77 *Salmonella* isolates from bovine Lymph Nodes (LN) and Ground Beef (GB).** Within each antibiotic class, AMR phenotypes are first indicated, followed by AMR genes. Non-susceptible phenotypes are indicated with "R" (resistance) and "I" (intermediate resistance), while blank cells correspond to susceptible isolates. For AMR genotypes, cells filled with the corresponding antibiotic class color indicate the gene is present. AMR genes in gray are not associated with any antibiotic class included in the AST panel. Point mutations and MDR phenotypes are presented on the rightmost columns. Isolates with black cells have mutations/MDR phenotypes and those with blank cells lack these features. Antibiotic abbreviations are as follows: amikacin (AMK), gentamycin (GEN), carbenicillin (CB), amoxicillin-clavulanic acid (AMC), cefotaxime (CTX),

ceftriaxone (CRO), cefepime (FEP), meropenem (MEM), ertapenem (ETP), imipenem (IMP), chloramphenicol (CHL), trimethoprim-sulfamethoxazole (SXT), ciprofloxacin (CIP), tetracycline (TET).

Genomic AMR profiling also identified the presence of several additional resistance genes that are not associated with any specific antimicrobial included in the AST panel. Particularly, those encoding resistance to fosfomycin (*fosA7.7*), as well as components of the multidrug and metal resistance-nodulation-division (RND) efflux complex (*mdsAB*). However, experimental isolates did not carry any known macrolide resistance gene (i. e. *mphAB*, *lnuF*, *ereA*, etc.).

Assembled genomes were also analyzed for point mutations associated with resistance. However, the occurrence of mutations was inconsistent with the observed phenotypes (Fig 1). For instance, all isolates carried multiple mutations in the quinolone resistance-determining region (QRDR), *gyrAB* and *parE* genes, regardless of whether they were susceptible or not to ciprofloxacin. Likewise, 100% isolates had mutations in *soxRS* genes, which confer MDR profiles [44]. Still, only 26% of isolates were classified as MDR. Conversely, mutations in *ramR* were strongly associated with the occurrence of MDR strains ($\chi^2$ = 17.7, P<0.0001).

Genomic analysis also revealed additional widespread mutations. Those of *pmrAB* genes, which are associated with colistin resistance, were present in all isolates. Likewise, mutations in the *acrB* gene, which have been reported to confer resistance to azithromycin in typhoidal *Salmonella* strains [45], were detected in most of our isolates (68/77). Below, a detailed description of the relationship between AMR phenotypes and genotypes will be presented for each tested antibiotic class, as well as for MDR strains.

## Aminoglycoside resistance

Aminoglycoside-resistant isolates carried genes encoding enzymatic inactivation mechanisms, such as phosphorylation (several *aph* alleles) and adenylation (*aadA2* gene). Comparative analysis with public *Salmonella* genomes revealed a strong association between aminoglycoside resistance and the isolation source ($\chi^2$ = 242.7, P<0.0001). Human, avian and bovine strains were more likely to carry aminoglycoside resistance genes (OR = 4.6, 95CI 3.5–6.0, P = 0.0001) compared to other sources. However, the resistance profile was similar across sources, with a predominance of *aadA* and *aph* genes over the *acc* alleles (Fig 3).

## Beta-lactam resistance

Experimental isolates harbored only the Ambler's class A beta-lactamase-encoding genes *blaCARB-2* and *blaTEM-1*. These enzymes confer resistance to all penicillins, as well as first-, second- and third-generation cephalosporins [46]. The *blaCARB-2* gene was the most abundant among isolates showing resistance to penicillins, especially in those of serovar Typhimurium (9/10). One isolate of serovar Anatum and one of monophasic Typhimurium (1,4,[5],12:i:-) also carried *blaTEM-1*, which encode another extended-spectrum beta-lactamase (ESBL) that hydrolyzes penicillins and first-generation cephalosporins [46]. A total of 11 non-susceptible isolates lacked any known ESBL-encoding gene. Strikingly, however, 10 of these isolates had *ramR* mutations, which are associated with resistance to several antibiotic classes, including penicillins.

Eight isolates were non-susceptible to 3GC and one to both 3GC and 4GC. However, none carried ESBL-encoding genes associated with these phenotypes. As observed with penicillins, mutations in *ramR* were associated with non-susceptibility to cephalosporins as 100% of non-susceptible isolates to 3GC/4GC had these mutations.

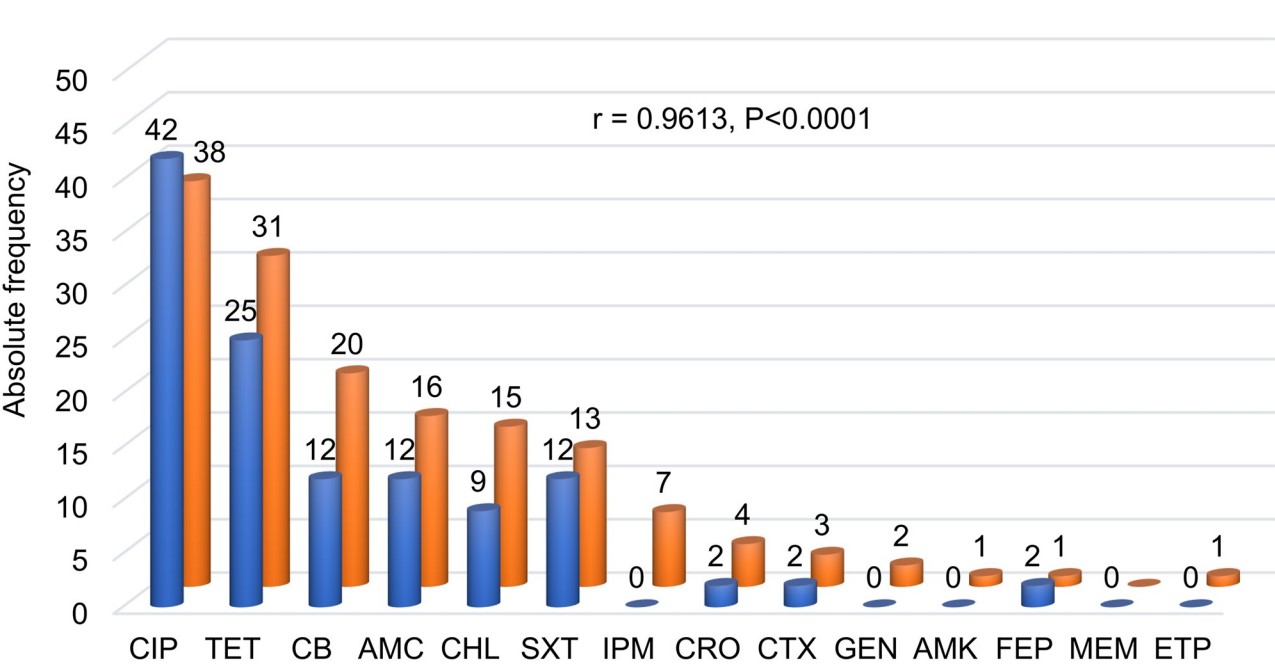

**Fig 2. Schematic representation of the overall correlation between predicted AMR genotypes and observed AMR phenotypes of 77 *Salmonella* isolates.** The figure shows the number of genotypically and phenotypically non-susceptible isolates to each tested antibiotic. Antibiotic abbreviations: ciprofloxacin (CIP), tetracycline (TET), carbenicillin (CB), amoxicillin-clavulanic acid (AMC), chloramphenicol (CHL), trimethoprim-sulfamethoxazole (SXT), imipenem (IMP), ceftriaxone (CRO), cefotaxime (CTX), gentamycin (GEN), amikacin (AMK), cefepime (FEP), meropenem (MEM), ertapenem (ETP).

Non-susceptibility to carbapenems was the least frequent among the tested antibiotic classes. No isolate resisted meropenem, while only one showed intermediate resistance to ertapenem and seven to imipenem. Again, no carbapenemase-encoding genes were found in the genome of any non-susceptible isolate. Interestingly, however, when using the ECOFF value of the inhibition zone (27 mm) set for meropenem in the European Union [32], 17 isolates (22%) were classified as non-wild type, which indicates some mechanisms of carbapenem resistance could be emerging in the population being studied.

Interestingly, isolates showing intermediate resistance to carbapenems carried a gene that has 99% sequence identity to the STM3737 gene of *Salmonella* Typhimurium (Uniprot accession Q8ZL44). This gene encodes a protein from the metallo-hydrolase-like-MBL-fold super family. A conserved domain search with the amino acid sequence of this gene at NCBI confirmed these findings (accession cl23716). We also found a positive hit (63% identity, E-value = $2.7 \times 10^{-115}$) with a *Serratia proteamaculans* beta-lactamase (genome GenBank accession CP000826.1) by running a BLAST analysis against the betalactamase database [47]. Whether this gene may confer reduced susceptibility to carbapenems remains unclear since it was present in the genome of all experimental isolates.

In relation to other public *Salmonella* genomes from Mexico, those from human, avian, and bovine species were the main source of beta-lactam-resistant *Salmonella*: $\chi^2 = 199.0$, P<0.0001, OR = 8.6, 95CI 5.8–12.7 (Fig 3). Overall, class-A ESBL-encoding genes (i. e. *bla-CARB*, *blaTEM*) were the most abundant, while *blaCTX-M* was predominant among avian isolates (17/193). Genes encoding class-C ESBLs (i. e. *blaCMY*) were also present in a small

Percentage of isolates carrying AMR alleles and having MDR genotypes

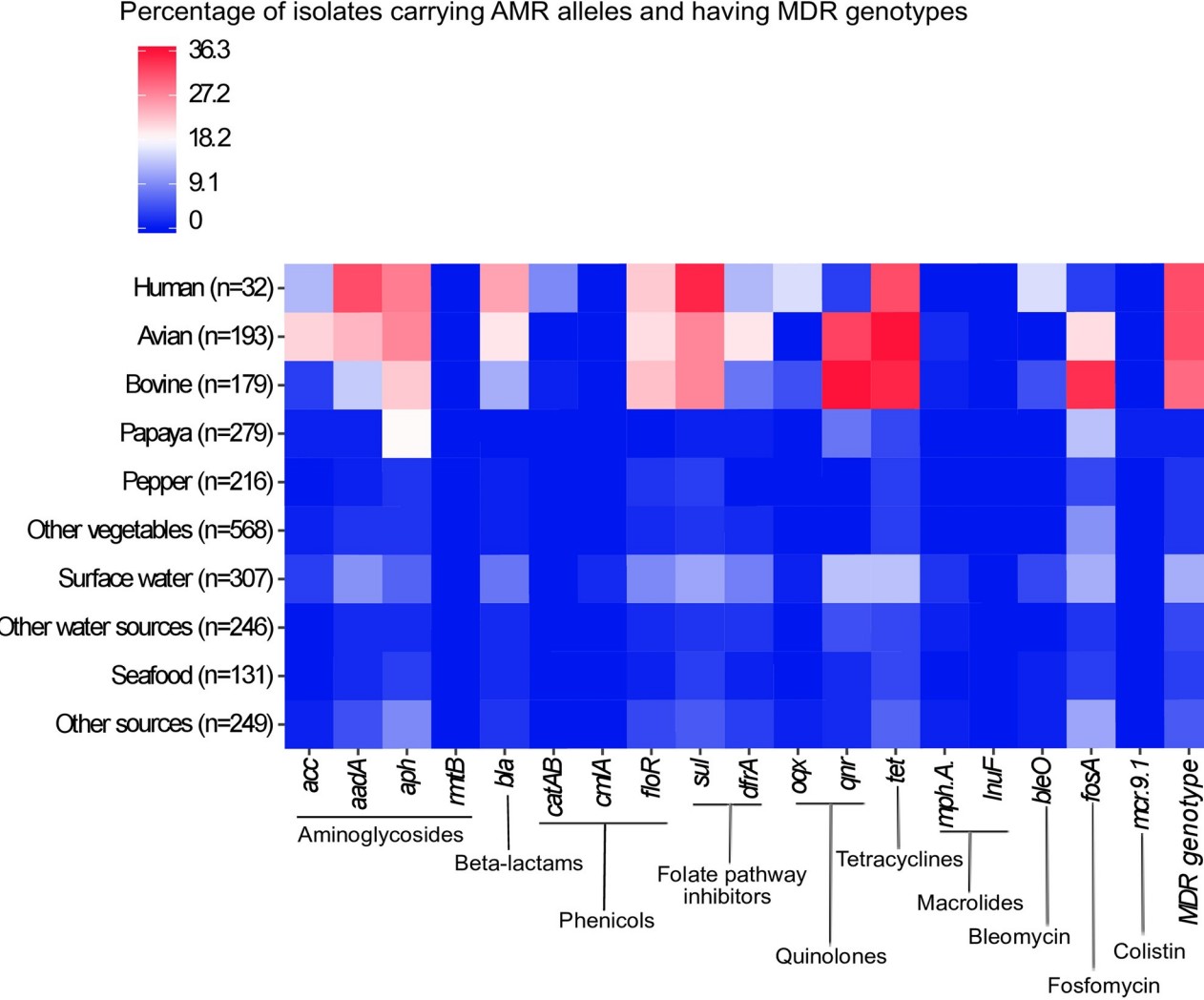

**Fig 3. Heatmap showing the AMR profile of public NTS isolated from major sources in Mexico.** AMR genes and the antibiotic class affected by them are indicated on the bottom. Refer to S2 File for accessions and metadata of isolates included in this analysis.

proportion of isolates from human (3/32), pepper (3/216), surface water (11/307), avian (1/193), and other sources (1/249). Only three human isolates carried a *blaOXA* gene, which encodes a class-D ESBL. Fortunately, class-B ESBL-encoding genes were not detected in the genome of the public isolates under study.

## Fluoroquinolone resistance

The main resistance mechanism we detected was that encoded by plasmid-mediated quinolone resistance (PMQR) genes, such as *qnrB19* and *qnrA1*, involved in quinolone target protection (DNA gyrase). This genomic feature, which confers low-level quinolone resistance, was strongly associated with the intermediate resistance to ciprofloxacin observed in our study ($\chi^2$ = 36.8, P<0.0001). As mentioned before, we did not find a statistical association between point-mutations in the QRDR and the observed phenotypes.

Among public *Salmonella* genomes, those from avian and bovine origin were the major source of PMQR genes ($\chi^2$ = 428.0, P<0.0001, OR = 11.6, 95CI 8.6–15.6). Over 30% of these

isolates harbored *qnr* alleles, while *oqx* ones were rarely found. Conversely, the proportion of PMQR-positive isolates was 18.8% in human isolates, with *oqx* alleles predominating over *qnr* ones. The frequency of PMQR genes in other sources was low (<5%), with slightly higher values in surface water (13.7%) and papaya (7.5%). Again, *qnr* alleles were the most abundant.

## Chloramphenicol resistance

Resistance to chloramphenicol in experimental isolates was mainly associated with efflux mechanisms (*floR* gene). This gene was present in most isolates of serovar Typhimurium (8/10), accounting for over 50% of chloramphenicol non-susceptible phenotypes. We detected a second resistance mechanism (antibiotic inactivation), encoded by a chloramphenicol acetyltransferase gene (*catA2*). However, this gene was present in just one isolate of serovar Give. Moreover, six non-susceptible isolates were predicted as genotypically susceptible. Again, *ramR* mutations were strongly associated with chloramphenicol resistance ($\chi^2$ = 8.1, P = 0.0045), OR = 21.1 95CI 1.2–368.4.

Among public NTS isolates, the proportion of phenicol-resistant genotypes was associated with the isolation source ($\chi^2$ = 252.4, P<0.0001). Once more, human, avian and bovine strains were more likely to carry phenicol resistance genes than isolates from any other source: OR = 9.4, 95CI 6.6–13.2. Still, the genomic AMR profile was similar across sources (Fig 3). Efflux factors (i. e. *floR*) predominated over enzymatic inactivation mechanisms (i. e. *cat* and *cml* alleles).

## Folate pathway inhibitors resistance

Regarding folate pathway inhibitors, the most abundant resistance mechanism among experimental isolates was that encoded by *sul* alleles (i. e. *sul1* and *sul2*), which were present in 11 isolates. Conversely, just one isolate carried the *dfrA12* gene along with the *sul1* gene. Overall, there was a discrete proportion of experimental isolates that resisted trimethoprim-sulfamethoxazole (16.9%).

The resistance profile of public genomes again revealed isolates collected from human, avian and bovine species were more likely to carry resistance genes against folate pathway inhibitors, as compared to those from other sources ($\chi^2$ = 272.9, P<0.0001), OR = 8.6 95CI 6.3–11.7 (Fig 3). The most common AMR allele was *sul*, which predominated over *dfrA*.

## Tetracycline resistance

Among experimental isolates, tetracycline resistance was mainly associated with the presence of genes encoding efflux mechanisms (i. e. *tetABCG*). Seven non-susceptible isolates did not carry any known tetracycline resistance gene. However, all these isolates carried *ramR* mutations, which are associated with MDR profiles involving tetracyclines and other antibiotic classes [48].

Public *Salmonella* genomes from human, avian and bovine species had the highest proportion (31.3, 35.8, and 34.6%, respectively) of tetracycline-resistance genotypes ($\chi^2$ = 327.9, P<0.0001), OR = 8.6 95CI 6.3–11.7 (Fig 3). Conversely, the frequency of *tet* alleles in isolates from other sources was discrete (3.2–13.7%). Regarding the genomic AMR profile, isolates from all sources carried at least one variant of the same efflux-mediated resistance determinant (*tetABCDGM*).

## MDR profiles

Among experimental isolates, MDR profiles were associated with the presence of a resistance island and, to a lesser extent, the occurrence of mutations in their genomes. For instance, most

serovar Typhimurium isolates (9/10) carried SGI1. This genomic island harbors a class-1 integron containing multiple gene cassettes (i. e. *aadA2*, *blaCARB-2*, *floR*, *sul1*, *tetG*), which explains the observed MDR phenotypes of these isolates (Fig 4). This feature seems common in *S. enterica* ser. Typhimurium circulating in Mexico. The analysis of the whole set of Typhimurium isolates from Mexico deposited at NCBI (n = 40) showed 52.5% of them have MDR genotypes. Among these, nearly 80% harbor multiple AMR genes associated with the ACSSuT phenotype. Refer to S3 File for accession numbers and AMR genotypes of this group of isolates.

As mentioned before, mutations in the *ramR* gene (Fig 1) were strongly associated with MDR phenotypes as well. In fact, the probability of finding MDR strains was 50.5 times higher (95CI 2.9–881.3) in isolates carrying *ramR* mutations than in those lacking them.

Interestingly, plasmids had a rather discrete contribution to AMR phenotypes in general (Table 2). Only one-third of the isolates being studied (n = 26) were predicted to carry seven plasmids. Of these, three were small plasmids harboring few replication-related genes. The most abundant plasmid was the *Salmonella* virulence plasmid (pSLT), which was detected in all serovar Typhimurium isolates. The pSLT is a hybrid plasmid that carries a class-1 integron with several AMR gene cassettes against beta-lactams (*blaTEM*), chloramphenicol (*catA*), aminoglycosides (*aadA1*, *strAB*), and folate pathway inhibitors (*sul1*, *sul2*, *dfrA1*). However, the integron carried by our serovar Typhimurium isolates had a genomic context matching that of SGI1 instead of pSLT (see Fig 4 and S1 Fig).

Among the plasmids detected, the pK245 has the strongest resistance profile. It carries a class-1 integron with a sole resistance cassette (*dfrA14*), as well as another seven AMR genes (*tetAR*, *sul2*, *strAB*, *catA2*, *blaSHV-2*) distributed across the plasmid sequence. However, it is unlikely that pK245 plasmid was present in that isolate. Only 30% of this plasmid aligned to the genome of the single isolate (a serovar Give strain) carrying its replicons. Moreover, this isolate carried a chromosomal class-1 integron with seven resistance cassettes in a genomic context different from that of pK245 (see S2 Fig).

Replicons of the resistance plasmid R64 were also found in the genomes of serovar Fresno isolates (n = 4), as well as one serovar Anatum isolate. This plasmid carries AMR genes against tetracyclines (*tetADR*) and aminoglycosides (*strAB*). Although more than 70% of R64 aligned to the assembled genomes, none of its AMR genes were found in the genomes of isolates carrying its replicons, except *tetA*, which was detected in the serovar Anatum isolate (see S3 Fig).

Finally, the monophasic Typhimurium isolate carried replicons of the pOLA52 plasmid. This hybrid plasmid harbors genes encoding a type IV secretion system, as well as AMR genes against quinolones (*oqxAB*) and beta-lactams (*blaTEM*). Although more than 70% of this plasmid aligned to the referred genome, only the *blaTEM* gene was also present in the isolate, while *oqxAB* genes were missing (see S4 Fig).

The analysis of public genomes showed human, avian and bovine isolates are the most important sources, among those studied, of MDR NTS genotypes ($\chi^2$ = 321.0, P<0.0001), OR = 9.7 95CI 7.2–13.1 (Fig 3). In the other matrices, the proportion of MDR genotypes was 5.6% or less, except for surface waters (11.7%).

Among the antimicrobials that were excluded from the AST panel, we observed only a relative abundance of fosfomycin resistance alleles (*fosA*). These were more frequently found among bovine (32.4%) and avian (20.7%) isolates ($\chi^2$ = 141.6, P<0.0001), OR = 3.7 95CI 2.9–5.0. In the other sources, the proportion of *fosA*-positive isolates was 13.3% or lower. Macrolide (*mph* and *lnuF*), and colistin (mcr-9.1) resistance genes were seldom found (0–2.6% across sources), while human isolates were the most significant source of bleomycin resistance alleles (*bleO*), as compared to other sources: $\chi^2$ = 94.0, P<0.0001, OR = 15.4 95CI 5.6–43.1.

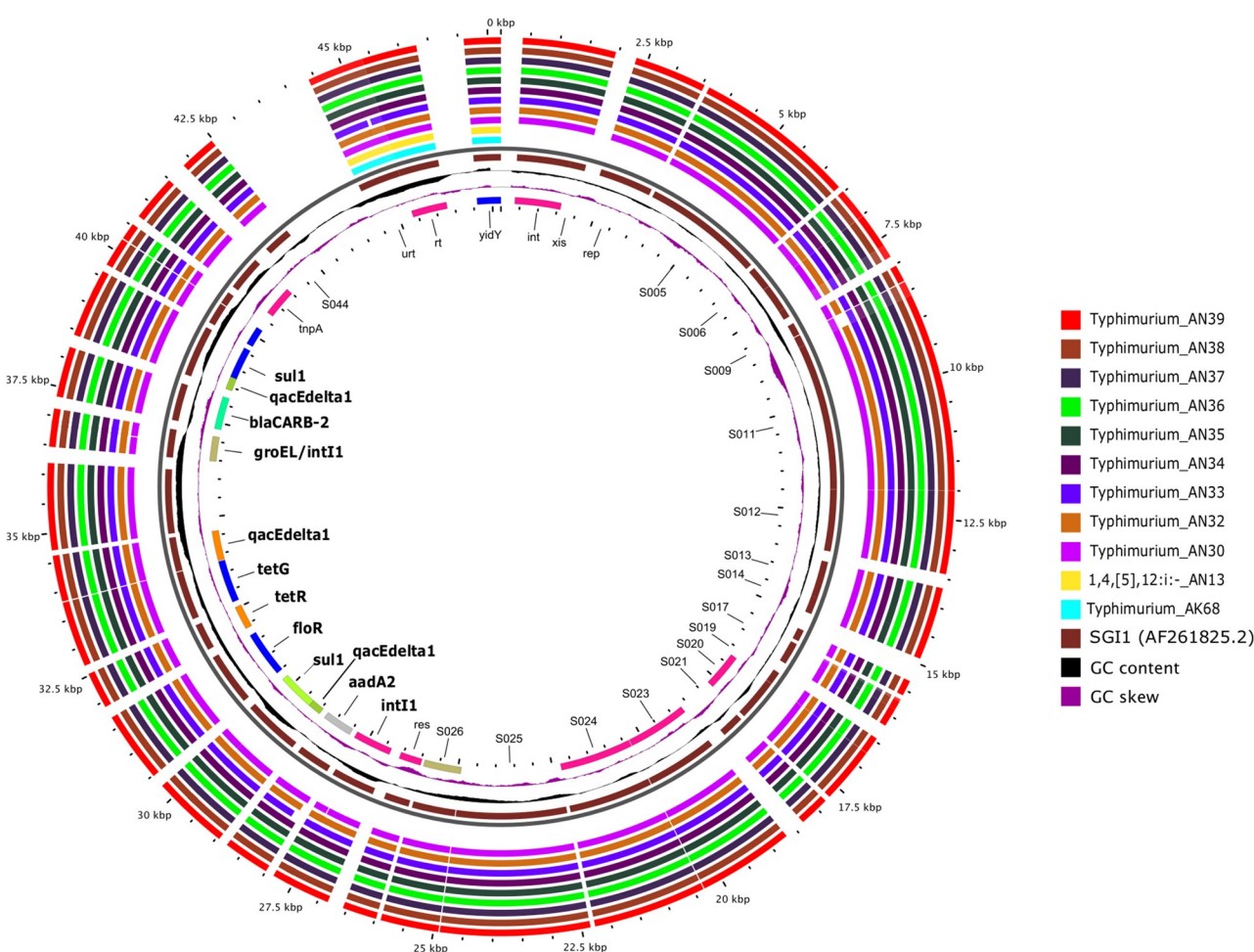

**Fig 4. BLAST atlas analysis of SGI1 of 10 serovar Typhimurium isolates and one monophasic Typhimurium.** The black slot corresponds to the backbone and the brown inner ring to the reference sequence. The most inner ring shows gene annotation, with AMR genes highlighted in bold text. The outer rings correspond to the serovars and isolate names indicated. Regions with shared synteny between genomes and reference sequence are filled with intense color while blank spaces indicate a lack of synteny. Refer to S1 File for accession numbers and metadata of isolates.

**Table 2. General features of the plasmids predicted per *Salmonella* serovar.**

| Plasmid (NCBI accession) | Size, bp | Incompatibility group | Plasmid AMR genes[1] | Serovar (n)[2] |
|---|---|---|---|---|
| pSLT (FN432031) | 117,047 | IncFIB(S) | ***aadA1***, *blaTEM*, *catA1*, *dfrA1*, *strAB*, ***sul1***, *sul2* | Typhimurium (10) |
| pK245 (DQ449578) | 98,264 | IncR | *aacC2*, *blaSHV2A*, ***catA2***, *dfrA14*, *strAB*, *sul2*, *tetD*, *tetR* | Give (1) |
| R64 (AP005147) | 120,826 | IncI1 | *tetACD*, *strAB* | Fresno (4) |
| | | | | Anatum (1) |
| pOLA52 (EU370913.1) | 51,602 | IncX1 | ***blaTEM***, *oqxAB* | 1,4,[5],12:i:- (1) |
| RSF1010 (M28829.1) | 8,684 | IncQ1 | — | 1,4,[5],12:i:- (1) |
| pBS512_2 (NC_010656.1) | 2,089 | — | — | Anatum (2) |
| | | | | London (4) |
| | | | | Reading (1) |
| pVCM01 (JX133088) | 1,981 | — | — | Anatum (2) |

[1]Only AMR genes in bold were present in both the plasmid and genome.

[2]Number of isolates of the same serovar carrying replicons of the plasmid.

## Discussion

Our results confirmed previous observations of widespread resistance to older antibiotics (i. e. tetracyclines, penicillins) among NTS from different sources. For instance, in Mexico, previous studies consistently report these phenotypes at high frequencies (up to 90% or higher) in beef isolates [12, 15, 17, 49–51]. Likewise, these AMR phenotypes are also common in NTS isolated in developed countries, a phenomenon that is considered driven by the use of these antimicrobials in food-producing animals [27].

Regarding tetracyclines, selective pressure has likely resulted in extensive acquisition of efflux mechanisms (*tetABCDGM* alleles), which are usually carried either in plasmids or in the chromosome of NTS [52]. This study also supports mutations could sustain this phenotype in isolates lacking *tet* alleles, providing evidence of convergent evolution. As observed here, 100% of isolates that did not carry *tet* alleles had *ramR* mutations. The *ramR* gene encodes a protein that represses the expression of *ramA* by binding to its promoter region. Certain mutations in *ramR*, such as those detected here (i. e. Y59H, M84I, E160D), affect the DNA binding affinity of RamR and reduce the repression of the *ramA* gene. This disruption leads to overexpression of *ramA*, which activates the major efflux-mediated AMR mechanism in Gram-negative bacteria: the *acrAB-tolC* RND multidrug efflux pump [53, 54]. This feature is a broad-spectrum efflux system that confers resistance to unrelated antibiotic classes such as tetracyclines, quinolones, phenicols and beta-lactams [55].

Strikingly, results showed that bovine and avian species seemed to be the most relevant source of tetracycline resistance NTS, as compared to other food-related sources. Tetracyclines are highly important antimicrobials [26] since they are among the few alternatives to treat human infections caused by *Brucella* spp., a pathogen that is associated with cattle. Hence, these findings are relevant from a public health perspective.

The rate of resistance to penicillins observed here was not as high as that reported recently in NTS isolated from beef in Mexico [14, 15, 17, 56]. However, this variation is likely associated with the region where isolates originated. Hence, the importance of cattle as a relevant source of penicillin-resistant NTS should not be discarded, as shown by the genomic comparison with other public isolates from Mexico conducted here.

In beta-lactam resistance, class-C and class-B ESBLs are leading concerns. Class-C ESBLs confer resistance to most beta-lactams (except carbapenems) and resist clavulanic acid, while class-B metallo-beta-lactamases confer the strongest resistance phenotype, involving all known beta-lactams and clavulanic acid [46]. Fortunately, resistance to 3GC, 4GC and carbapenems was rare among experimental isolates, which is in agreement with recent studies [9]. Apparently, food isolates in Mexico, as well as in the US and Europe [27], are not a significant source of NTS resistant to 3GC/4GC and carbapenems. Still, we observed some strains (1–9%) showed non-susceptibility to 3GC, 4GC and carbapenems, despite lacking ESBL-encoding genes. In this regard, the most plausible explanation is the presence of *ramR* mutations, given their strong association with beta-lactam resistance observed here.

The widespread distribution of *ramR* mutations in the isolates being studied may be the result of selective pressure. In Mexico, 3GC used in human medicine (i. e. cefotaxime, ceftriaxone) are also approved, besides ceftiofur, to treat several bovine diseases [28]. Ceftiofur is associated with the emergence of ceftriaxone resistance among livestock and poultry NTS in the United States [27]. Hence, it is important to revise the approval of both ceftiofur and other 3GC for veterinary use in Mexico. It has been reported that restrictions (voluntary or law-enforced) in 3GC use in the United States and Canada have helped reduce 3GC resistance among NTS associated with animals [27].

The moderate proportion of isolates that exceeded the EUCAST's ECOFF value for meropenem (22%), which is a last-resort antibiotic, also highlights the importance of continuous surveillance of NTS from foods. We did not observe an association between any predicted genetic determinant and reduced susceptibility to meropenem. Hence, further research and continued surveillance are needed in this area.

We also observed intermediate resistance to ciprofloxacin among experimental isolates, a phenomenon associated with the presence of PMQR genes. Particularly, the *qnrB19* allele, which was present in over 50% of our experimental isolates, has also been reported as the predominant among NTS strains from the United States [25]. Recent studies from Mexico have also documented high rates (36–44%) of decreased susceptibility to ciprofloxacin in beef NTS isolates [12], although these authors did not determine AMR genotypes. Moreover, a previous study by our research group documented *qnr* and *oqx* alleles were widely distributed among NTS isolated from cattle feces, carcasses and ground beef [17]. Furthermore, genomic AMR profiling of public NTS isolates showed *qnr* alleles are widely disseminated among bovine and avian isolates. Conversely, they are less frequently found in isolates from other sources, including those of human origin.

In the context of Mexico's animal production, acquisition and conservation of PMQR genes in NTS is likely the result of selective pressure associated with the use of quinolones (i. e. enrofloxacin). These are approved to treat several diseases in cattle and poultry [28] and the fact that PMQR genes are carried in plasmids increases the risk of their dissemination under positive selective pressure.

Many studies report quinolone resistance is rare in NTS isolated from cattle [3, 57]. Hence, it has been suggested that the use of these drugs in cattle could not be linked to the emergence of quinolone resistance among cattle NTS isolates [1]. However, PMQR genes confer low-level quinolone resistance, which goes undetected when using CLSI breakpoints [58]. Therefore, we believe that the role of PMQR genes as a contributing factor to quinolone resistance in NTS from cattle should not be minimized. Especially, considering PMQR genes have been associated with increasing resistance to quinolones in NTS isolated from foods [59, 60]. In summary, cattle and poultry are relevant reservoirs of low-level quinolone-resistant NTS. However, to what extent PMQR genes could lead to the emergence of clinical fluoroquinolone resistance is yet to be determined.

Regarding aminoglycoside resistance, although the number of non-susceptible isolates in the experiment was low, the CLSI guidelines emphasize these antimicrobials may appear active *in vitro* but are not effective clinically against *Salmonella* and, thus, susceptible isolates should not be reported as such [30]. This situation may explain why isolates carrying *aadA* and *aph* alleles exhibited susceptible phenotypes, a phenomenon observed in previous experiments [17]. However, comparative genomics of public NTS isolated in Mexico showed poultry and cattle are as important as humans as a source of aminoglycoside resistant genes.

Resistance to chloramphenicol and trimethoprim-sulfamethoxazole was frequently observed among experimental isolates (approximately 17 and 20%, respectively). Strains showing these phenotypes carried AMR genes (i. e. *cat*, *flo*, *sul*, and *dfrA* alleles) against both antibiotics. In the case of chloramphenicol, *ramR* mutations, which confer phenicol resistance [48], seemed to play a role as well.

The selective pressure exerted by the use of these drugs in food-producing animals may explain these findings. For instance, trimethoprim-sulfamethoxazole is approved as a wide spectrum antibiotic to treat all sorts of bacterial infections in livestock and poultry in Mexico [28]. Likewise, although chloramphenicol is no longer approved for these purposes, there are other phenicols (i. e. florfenicol) registered. This situation is likely the reason why phenicol resistance is consistently reported in NTS from animal foods in Mexico, in proportions that

vary from moderate (16–23%) [12, 15, 51] to high (>90%) [14, 56]. Our comparative genomics further supported this analysis since *floR*, which confers resistance to both phenicols [61], was the dominant resistance factor harbored by NTS from food-related sources and humans in Mexico.

Regarding MDR profiles, it is interesting to note the higher frequency of MDR isolates in ground beef compared to lymph nodes. Especially, considering lymph nodes are known to be major contributors to NTS contamination in ground beef [62]. These findings support previous Mexican studies documenting a high rate of MDR phenotypes (~30–70%) in NTS from ground beef [12, 13, 15, 17], as well as a low proportion of MDR isolates (~8–13%) from lymph nodes [8, 63]. Perhaps, the fact that isolates collected from lymph nodes survive within host cells [64] and thus, are not exposed to antibiotics, results in a weaker antimicrobial selective pressure. Conversely, ground beef isolates originating from sources other than lymph nodes are more likely exposed to antibiotics, such as those of fecal origin. Hence, they exhibit stronger AMR profiles. Our study further supports this hypothesis since ground beef and lymph nodes from the same carcass were analyzed separately here. Apparently, lymph nodes are not the source of a relevant proportion of strains circulating in ground beef. At least, not in the context of the Mexican beef industry, where carcass fecal contamination seems also relevant. In that sense, there are reports showing higher carcass NTS contamination rates in Mexico (6–18%) [51, 65, 66] compared to developed countries (<1%) [67, 68]. Likewise, recent studies report NTS prevalence in ground beef of 15 to over 70% in Mexico [13, 15, 65], while that observed in Ireland, Belgium and the USA is below 5% [67, 69, 70]. Moreover, phylogenomic analysis has documented the fecal origin of some NTS isolates found in beef carcasses, cuts and ground beef in Mexico [41].

Given these facts, it is reasonable to think carcass fecal contamination is a more significant source of MDR *Salmonella* in ground beef compared to lymph nodes. This analysis is consistent with conditions prevailing in the beef industry of LMIC. In these nations, there are limited controls on the use of antimicrobials in animal production, as well as poor carcass fecal contamination control at slaughter. Still, NTS also travels from the intestine to lymph nodes. Hence, further research is needed to better understand the possible influence of differing ecological conditions across bovine isolation sources on NTS AMR profile.

Certain *Salmonella* STs are associated with MDR phenotypes. Among our isolates, those of serovar Typhimurium (ST-19), monophasic Typhimurium (ST-34) and Kentucky (ST-198) are the most relevant in this respect [21–23]. With few exceptions, most of these strains exhibited MDR phenotypes and genotypes, confirming the acquisition of AMR determinants is a hallmark of epidemiologically relevant NTS strains.

Among serovar Typhimurium isolates, SGI1 was the dominant genomic feature associated with MDR phenotypes. The SGI1 contains a class-1 integron and multiple AMR gene cassettes (*aadA2*, *floR*, *tetG*, *blaCARB-2*, *sul1*) conferring resistance to ampicillin, chloramphenicol, streptomycin, sulfonamide, and tetracycline (known as the ACSSuT phenotype) [71]. This penta-resistance profile is typical of MDR *S. enterica* ser. Typhimurium DT104 carrying SGI1 [23] and was similar to that observed here. Although SGI1 is not self-mobilizable, donor cells harboring conjugative plasmids may transfer it to other hosts [72]. In fact, SGI1 has been recognized as a key factor for the rapid dissemination of strains [73].

Unfortunately, although serovar Typhimurium is the most frequently found in NTS circulating in foods and clinical cases in Mexico, it has been poorly characterized in terms of its genomic AMR profile [9]. According to that review, the SGI1 associated ACSSuT phenotype has been reported in a modest proportion of MDR isolates (9.5%) collected from clinical cases, chicken, pork, and beef between 2006 and 2013 [9]. Analysis of public serovar Typhimurium isolates from Mexico deposited at NCBI (n = 40) showed 52.5% of them carry AMR genes

supporting DT104-like or even stronger phenotypes. Nonetheless, further research is needed to establish if the high rate of DT104-like isolates observed here is a local dissemination phenomenon. Especially, considering all of our MDR Typhimurium isolates originated from the same batch of carcasses and were collected on the same date. In this respect, serovar and ST seem more relevant, in terms of AMR profiles, than season or date of collection. As observed here, serovar Typhimurium isolates represented 40% of MDR phenotypes and they were all isolated in the autumn of 2018. Undoubtedly, it resulted in a strong association between the season/year of collection and the rate of MDR phenotypes among NTS isolates.

Interestingly, other MDR isolates lacked acquired AMR genes or genomic features (i. e. resistance islands, integrons) sustaining these phenotypes. However, all MDR isolates had *ramR* mutations, which have been observed to correlate well with MDR phenotypes [48, 74]. Still, the molecular mechanisms associated with *ramR* mutations and AMR in *Salmonella* are complex and have not been fully deciphered. Recent studies have documented the differential impact of mutations in the *ramRA* regulatory region on *ramA* transcription and AMR profiles of NTS [74]. This phenomenon could be the reason why some isolates with the same *ramR* mutations were pansusceptible, mono- or bi-resistant, granting the need for further research in this area.

Mutations in the *soxRS* regulon, which were present in 100% of our experimental isolates, also confer MDR phenotypes in Gram-negative bacteria [75]. However, *soxRS* genes are activated only by the host's immune attack and inflammation [44]. This evidence is consistent with the lack of association between MDR phenotypes and the widespread *soxRS* mutations observed here. Nonetheless, these strains might exhibit stronger AMR profiles in clinical settings, which highlights the importance of preventing NTS dissemination along the food chain.

Despite plasmids play a major role in the acquisition and dissemination of bacterial AMR [76], they had a limited contribution to the MDR phenotypes observed in this study. A rational explanation for these findings is the presence of other genomic features, such as SGI1 and mutations, fulfilling the same function. In this context, plasmid-borne AMR genes likely provide no fitness advantages, leading to their eventual excision. For instance, the class-1 integron present in the most abundant plasmid (pSLT), carried by serovar Typhimurium isolates (n = 10), lacked most of its AMR gene cassettes. However, most of these isolates carried SGI1, which harbors a similar integron. In integrons, AMR gene cassettes are integrated by the integron integrase and expressed via the integron promoter, and thus immediately subjected to natural selection [77]. However, this process is reversible as AMR gene cassettes may also be excised if they do not confer fitness advantages.

Macrolide resistance genes were not detected in experimental isolates and were infrequent (<1% overall) among NTS public isolates. These findings support previous studies documenting macrolide resistance phenotypes are rare among NTS isolated from foods in Mexico [9]. Nevertheless, nearly 90% of experimental isolates carried mutations in the *acrB* gene. These mutations have been associated with macrolide resistance in typhoidal strains [45]. Since we excluded macrolides in the AST, it was not possible to determine whether the *acrB* mutations were associated with macrolide resistance. To avoid this limitation, future studies should include macrolide antibiotics in the AST panel.

Bovine and avian isolates were also the most relevant source of AMR genes against fosfomycin. This antibiotic is considered critically important and is frequently used to treat urinary tract infections in humans [30]. Nevertheless, it is also approved in Mexico to treat cattle, pigs and poultry [28], which seems to be the reason for the relative abundance of *fosA* alleles among bovine and avian isolates observed here.

Overall, genomic AMR profiling of public isolates showed that there is some obvious dissemination of AMR genes in the environment. Although at lower frequencies, there are AMR

alleles against most antimicrobial classes in isolates from ecological niches where exposure to antibiotics is not intense (i. e. vegetables, water). Nonetheless, this hypothesis needs to be confirmed with further studies.

Taken together, our results illustrate the diversity of factors influencing the AMR profile of NTS across ecological niches. Particularly, the acquisition of resistance islands, plasmids, or mutations may confer advantageous phenotypes under antimicrobial selective pressure. This situation may lead to the rapid emergence and spread of MDR strains, as previously documented for *Salmonella* Typhimurium DT104 and Kentucky ST-198 [22, 23]. Likewise, our comparative analysis showed NTS strains isolated from cattle and poultry have strong AMR genotypes, which are similar to that of human clinical isolates. These findings suggest that food production practices are likely contributing to selection of AMR bacterial pathogens. Hence, it is vital to improve NTS control in apparently healthy animals to prevent its dissemination along the food chain and, consequently, human exposure to MDR strains. Moreover, we believe that attaining significant improvements in AMR meat safety may require the identification and removal (or treatment) of product harboring MDR NTS, instead of screening for isolates showing resistance to individual antimicrobial classes. Such measures do not seem realistic, in the context of the meat industry of many countries, where testing practices are limited to *Salmonella* spp. isolation and confirmation. However, most nations have embraced the WHO global action plan on AMR. Thus, they should eventually set up existing technologies, such as WGS, which provide the shortest path to accomplish these goals.

## Supporting information

**S1 File. Metadata of experimental isolates.**
(XLSX)

**S2 File. Metadata of public Salmonella isolates included in this study.**
(XLSX)

**S3 File. Metadata of public Typhimurium isolates included in this study.**
(XLSX)

**S1 Fig. BLAST atlas of plasmid pSLT.**
(PDF)

**S2 Fig. BLAST atlas of plasmid pK245.**
(PDF)

**S3 Fig. BLAST atlas of plasmid R64.**
(PDF)

**S4 Fig. BLAST atlas of plasmid pOLA52.**
(PDF)

## Acknowledgments

The authors appreciate the technical assistance and training on whole genome sequencing provided by the staff of the National Reference Center for Pesticides and Contaminants belonging to the General Directorate of Agri-Food, Aquaculture and Fisheries Safety of the national Service for Agri-food Health, Safety and Quality. We are also grateful for the support of laboratory technicians, as well as graduate and social service students from the Faculty of Veterinary Medicine, National Autonomous University of Mexico, in field sampling and laboratory analyses.

## Author Contributions

**Conceptualization:** Enrique Jesús Delgado-Suárez, Orbelín Soberanis-Ramos, Rubén Danilo Méndez-Medina, María Salud Rubio-Lozano.

**Data curation:** Tania Palós-Guitérrez, Francisco Alejandro Ruíz-López, Cindy Fabiola Hernández Pérez.

**Formal analysis:** Enrique Jesús Delgado-Suárez, Tania Palós-Guitérrez, Cindy Fabiola Hernández Pérez, Nayarit Emérita Ballesteros-Nova.

**Funding acquisition:** Orbelín Soberanis-Ramos, Marc W. Allard, María Salud Rubio-Lozano.

**Methodology:** Enrique Jesús Delgado-Suárez, Francisco Alejandro Ruíz-López, Cindy Fabiola Hernández Pérez.

**Project administration:** Orbelín Soberanis-Ramos, María Salud Rubio-Lozano.

**Resources:** Francisco Alejandro Ruíz-López, Orbelín Soberanis-Ramos, María Salud Rubio-Lozano.

**Software:** Enrique Jesús Delgado-Suárez, Cindy Fabiola Hernández Pérez, Nayarit Emérita Ballesteros-Nova.

**Supervision:** Enrique Jesús Delgado-Suárez, Rubén Danilo Méndez-Medina, Marc W. Allard, María Salud Rubio-Lozano.

**Visualization:** Enrique Jesús Delgado-Suárez, Tania Palós-Guitérrez.

**Writing – original draft:** Enrique Jesús Delgado-Suárez, Tania Palós-Guitérrez.

**Writing – review & editing:** Enrique Jesús Delgado-Suárez, Tania Palós-Guitérrez, Francisco Alejandro Ruíz-López, Cindy Fabiola Hernández Pérez, Nayarit Emérita Ballesteros-Nova, Orbelín Soberanis-Ramos, Rubén Danilo Méndez-Medina, Marc W. Allard, María Salud Rubio-Lozano.

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
