## [Decision Letter · Decision Letter 0]

23 Dec 2020

PONE-D-20-36934

Genomic surveillance of antimicrobial resistance shows cattle are a moderate source of multi-drug resistant non-typhoidal Salmonella in Mexico

PLOS ONE

Dear Dr. Delgado-Suárez,

Thank you for submitting your manuscript to PLOS ONE. After careful consideration, we feel that it has merit but does not fully meet PLOS ONE’s publication criteria as it currently stands. Therefore, we invite you to submit a revised version of the manuscript that addresses the points raised during the review process.

A number of clarifications and explanations are needed including in illustrations.

We look forward to receiving your revised manuscript.

Kind regards,

Iddya Karunasagar

Academic Editor

PLOS ONE

Journal Requirements:

2. We note that you are reporting an analysis of a microarray, next-generation sequencing, or deep sequencing data set. PLOS requires that authors comply with field-specific standards for preparation, recording, and deposition of data in repositories appropriate to their field. Please upload these data to a stable, public repository (such as ArrayExpress, Gene Expression Omnibus (GEO), DNA Data Bank of Japan (DDBJ), NCBI GenBank, NCBI Sequence Read Archive, or EMBL Nucleotide Sequence Database (ENA)). In your revised cover letter, please provide the relevant accession numbers that may be used to access these data. For a full list of recommended repositories, see http://journals.plos.org/plosone/s/data-availability#loc-omics or http://journals.plos.org/plosone/s/data-availability#loc-sequencing

Additional Editor Comments:

The reviewers have pointed out number of gaps in the manuscript. Clarifications and explanations are needed in all sections including methodology, results, discussion and data presentations in Tables and Figures. Please revise the manuscript considering all reviewer comments point by point.

Reviewers' comments:

Reviewer's Responses to Questions

**Comments to the Author**

1. Is the manuscript technically sound, and do the data support the conclusions?

Reviewer #1: Partly

2. Has the statistical analysis been performed appropriately and rigorously? 

Reviewer #1: No

3. Have the authors made all data underlying the findings in their manuscript fully available?

Reviewer #1: No

4. Is the manuscript presented in an intelligible fashion and written in standard English?

Reviewer #1: Yes

5. Review Comments to the Author

Reviewer #1: In this manuscript, the authors presented their phenotypic and genotypic AMR findings in NTS isolates derived from bovine lymph nodes and ground beef products in Mexico. They further compare their genotypic AMR findings with publicly available NTS genomes originated from Mexico. Overall, it is an interesting and valuable study that demonstrates the genomic similarities regarding the AMR profile between humans and bovine origin isolates in Mexico. However, I would suggest a major revision before publication. Please find more detailed comments below.

Overall: First of all, authors were able to find 2400 publicly available Salmonella isolates at NCBI which 1,714 were from a known source, thus included in the study. Authors should include other sources that have the major number of isolates clustered within (for example avian source represented with BioProject PRJNA480281) to avoid selection bias while measuring “ source-related association with AMR Salmonella”. Please also help me to understand the “unknown source” criteria that yield exclusion of the isolates from this study. Please kindly provide your search criteria for the isolates. In addition, there are several major sources (e.g., papaya, pepper, river, canal) that were classified under vegetables or the aquatic environment in this study. Authors should count for any major source separately in the analyses to increase resolution. Please also consider adding the “unknown” source related data as “other sources”

Authors should be able to report MLSTs of their isolates since they have already had WGS data. I believe this is highly important since AMR profiles and sub-serotypes are highly associated.

Another important aspect is the specific source of the 77 isolates should be provided along with the date of collection. These features need to be considered while driving any conclusion related to AMR profile and “source”. As a reviewer, I would like to see if the major group of Typhimurium isolates collected on 9/18/18 that show highly similar genotypic and phenotypic AMR profiles were all from the same batch of samples, city, or producer. Please clarify also in the manuscript and provide data.

Abstract: Authors did not talk about 3GC in the abstract. Please mention the function/importance of ramR gene for MDR. The authors did not discuss their findings for high levels of amoxicillin-clavulanic acid-resistant isolates in the absence of AMR genes related to this phenotype in the manuscript.

Lines 100-102 Authors should explain why they choose these individual lists of antibiotics while there are many other antibiotics/classes listed in WHO. AMR in Salmonella is especially important when resistance emerges for the antibiotics that are used to treat Salmonella infections in humans. Macrolides (e.g., azithromycin) are one of the important antibiotics to treat Salmonella infections and this antibiotic was not included in this study. How would the authors explain their selection of antibiotics is unclear in the manuscript. Please also provide the class information of selected antibiotics either in the manuscript or table/figures.

Line 133- Please describe which parameter was used to detect the “poor quality reads used for trimming criteria

Line 136- Please provide the version of the Spade tool. Did the authors conduct a quality check for their assemblies?

Line 147- Please refer to the human isolates as “human clinical cases” in related figures, tables, or manuscript as referred to in the line.

Line-149- Please provide the search term used to find these isolates and the criteria used. So, the readers can reach the same data analyzed in this study.

Line 151- Please provide if the AMRFinderPlus database and program that was mentioned is the same one that is used in this analysis. Please provide the information of the data analyzed for this study that was scanned using the same database and version of the AMRFinderPlus. Since these results are prone to change as the AMR database and software are updated, authors must confirm that they compare the outcomes of WGS data using the same tools for all isolates included in the manuscript.

Line 152- Please provide the methodology for serotyping

Line 166- Please explain how the AMR profiles of each plasmid were determined.

Line 171- Why the authors select the %70 thresholds?

Line 175-180 These lines are misleading. There is no phenotype data included in the analysis of publicly available data. Please modify it accordingly.

Line 181- Even though the statistical language is sometimes used loosely in this regard, the authors performed analyses of association - not correlation - between phenotype and genotype and they will probably wish to change the wording to reflect that important difference.

Line 512-513 I would suggest authors review their statement related to WGS is being a better tool to monitor aminoglycoside resistance as compared to AST. The area of aminoglycoside resistance is difficult and evolving and interpretations are difficult and changing. Nonetheless, there is apparent cryptic genes aac(6’) and other factors such as breakpoints set for resistance for which phenotype and genotype seem disconnected. This disagreement does not infer that WGS is better, assuming that in at least some cases the naming conventions for the genes themselves might be based on flawed original experimentation, or else based originally in another genus or species.

Line 520 Please provide how many isolates had ramR mutations and did not harbor AMR genes but were found phenotypically resistant to chloramphenicol to support this statement.

Line 538-577 Authors may revise their justification related to the MDR presence in lymph node and ground beef as fecal origin Salmonella may travel from intestine to LN via payer’s patches, and transmission of environmental Salmonella may most likely occur via fly bites in the feedlots. Most importantly, lymph nodes are the most likely source of Salmonella found in ground beef products and contamination can occur via the fat trimming process.

Table 1 – Please explain “C”

Table 2- is confusing. Please consider reporting the point mutations and observed phenotype observed in 77 isolates that were not related to a defined AMR gene.

Table 3- how did the author screen the plasmidal content? Please include in the methodology

Figure 1- This table is hard to read. Authors may consider placing the class of phenotypic and genotypic resistance data next to each other for each class. I believe the qacEDelta1 is irrelevant with this research and is not shown on the AMRFinderPlus outcome of the isolates based on my research for given SAM IDs. Please help me to understand how this gene was found and why it was related to this study. Please also add ramR column here since the relationship between the mutation and phenotype has been analyzed and discussed.

Figure 2- Please provide the number of isolate information for each antibiotic. Even though the statistical language is sometimes used loosely in this regard, the authors performed analyses of association - not correlation - between phenotype and genotype and they will probably wish to change the wording to reflect that important difference.

Please consider using singular for sources. I would suggest revising the presentation of the source of isolates such as (e.g. human, bovine). Are these isolates have all genes listed in the conferring header (e.g., tetABCDGM) or at least one of them? Please clarify and add a footnote as needed.

S1 Table- please revise the number of sample size in the tab. I would suggest removing the collection date from Table 1 since it was not discussed in the manuscript. Please also provide the date of isolation and if possible the information about the source of isolates (if they were collected from the same city/abattoir etc.) of your isolates in S1. I would also suggest providing the phenotypic data in S1 along with the ramR mutation. There are duplicated gene names (.e.g., sul1), please correct. Please include the AMR related genes and mutations and isolate names as shown in Table 1 as they are not matching with supplemental material (the last column in Table 1 is missing. I would also suggest using the same isolate names consistent in the manuscript and related data. Many cells in the spreadsheet are missing, please revise. There are grammatical issues that need to be corrected. Authors need to revise AMR gene names and make sure they are presented correctly and consistently in both the body manuscript and related data.

S2 Table- Please correct the name on the tab. There are “environment or environmental sample” related isolates classified as the aquatic environment. Please revise to verify these isolates are actually from an aquatic source.

S3 Table- Please revise the AMR genotype column as there are duplicates, and information not relevant to AMR genes

Reference: Overall- References are not standardized, please correct the inconsistency observed with the lower- and upper-case use. Please revise all the links provided (e;g., URL of Ref 44 is not working) and provide accession dates for each URL. Please also italicize the spp names as needed.

6. PLOS authors have the option to publish the peer review history of their article (what does this mean?). If published, this will include your full peer review and any attached files.

Reviewer #1: No

---

## [Author Response · Author response to Decision Letter 0]

23 Feb 2021

The authors thank the reviewer and the academic editor for their valuable comments and suggestions that helped improve our manuscript. Below we list each comment raised during the reviewing process, followed by the authors’ responses.

Academic Editor Comments (AEC)

AEC1. Please ensure that your manuscript meets PLOS ONE's style requirements, including those for file naming. The PLOS ONE style templates can be found at

Authors’ Response (AR)1. We carefully reviewed the style requirements to make sure the manuscript meets them. Please, refer to the revised manuscript.

AEC2. We note that you are reporting an analysis of a microarray, next-generation sequencing, or deep sequencing data set. PLOS requires that authors comply with field-specific standards for preparation, recording, and deposition of data in repositories appropriate to their field. Please upload these data to a stable, public repository (such as ArrayExpress, Gene Expression Omnibus (GEO), DNA Data Bank of Japan (DDBJ), NCBI GenBank, NCBI Sequence Read Archive, or EMBL Nucleotide Sequence Database (ENA)). In your revised cover letter, please provide the relevant accession numbers that may be used to access these data. For a full list of recommended repositories, see http://journals.plos.org/plosone/s/data-availability#loc-omics or http://journals.plos.org/plosone/s/data-availability#loc-sequencing

AR2. The manuscript complies with this requirement. Please, refer to L121-131 (methods section) of the original manuscript. We declared raw sequences are publicly available at NCBI and provided the accession numbers and metadata in supplementary S1 Table. We also included the doi number for the two laboratory procedures that are citable and were uploaded at protocols.io.

AEC3. The reviewers have pointed out number of gaps in the manuscript. Clarifications and explanations are needed in all sections including methodology, results, discussion and data presentations in Tables and Figures. Please revise the manuscript considering all reviewer comments point by point.

AR3. The authors appreciate the time invested by the reviewers to make a very detailed and thorough revision of our manuscript. Comments will be taken into consideration to improve the paper and will be responded point by point.

Reviewer Comments (RC)

RC1. In this manuscript, the authors presented their phenotypic and genotypic AMR findings in NTS isolates derived from bovine lymph nodes and ground beef products in Mexico. They further compare their genotypic AMR findings with publicly available NTS genomes originated from Mexico. Overall, it is an interesting and valuable study that demonstrates the genomic similarities regarding the AMR profile between humans and bovine origin isolates in Mexico. However, I would suggest a major revision before publication. Please find more detailed comments below.

AR1. We appreciate these comments and will address every observation to improve our manuscript.

RC2. Overall: First of all, authors were able to find 2400 publicly available Salmonella isolates at NCBI which 1,714 were from a known source, thus included in the study. Authors should include other sources that have the major number of isolates clustered within (for example avian source represented with BioProject PRJNA480281) to avoid selection bias while measuring “ source-related association with AMR Salmonella”. Please also help me to understand the “unknown source” criteria that yield exclusion of the isolates from this study. Please kindly provide your search criteria for the isolates. In addition, there are several major sources (e.g., papaya, pepper, river, canal) that were classified under vegetables or the aquatic environment in this study. Authors should count for any major source separately in the analyses to increase resolution. Please also consider adding the “unknown” source related data as “other sources”. 

AR2. We only used two selection criteria: location (isolates from Mexico), and isolation source (the record should have a declared, unambiguous isolation source). Isolates of “unknown source” are those for which the isolation source was missing or ambiguous (i. e. sponge, swab, meat, product, etc.) in the record. We were interested in analyzing the AMR genomic profile of bovine isolates in the context of the bigger Salmonella population circulating in other sources. Our aim was to avoid making reports that are too general in nature. Instead, we attempted to be as specific as possible, to be able to answer specific relevant questions, such as the following: 

 Are cattle a relevant source of AMR-Salmonella? Are they as relevant as other food-related sources and clinical cases?

 What are the dominant genetic determinants associated with resistance to the studied antimicrobial classes?

 How disseminated are these AMR genes within and across sources? How the situation compares with that in other countries/regions?

 What factors are likely favoring AMR spread? What measures could be implemented to contain it?

The authors recognize committing several mistakes while conforming groups of isolates with the Excel file downloaded from NCBI. We overlooked the lack of uniformity in the records and relied on the “Filter” option of Excel to identify relevant groups. During this process, we missed all avian isolates, as well as some isolates from mostly every other source. We thank you for the detailed revision of the manuscript. In the updated version, we double checked the whole database to make sure there are no mistakes. Moreover, we agree to include all isolates from Mexico and thus, we created a new category named “Other sources, n=249”. This group includes isolates from sources with very few records, as well as those with ambiguous or unreported isolation source. Overall, the whole database includes 2,405 records now.

Regarding the comment on the need to break down categories into some major sources (i. e. papaya, pepper, river, etc.), we did the analysis separately before submitting the manuscript. While doing so, we observed there were minor differences within the vegetable and aquatic environment groups. So, we decided to analyze these categories as a whole for we did not see there was room for any resolution improvement. However, we decided to accept this comment and make the following adjustments:

The number of categories increased from 5 to 10:

1. Human, n=37

2. Bovine, n=179

3. Seafood, n=131. This category was not split since there was no variation in the AMR profile of isolates, regardless of the isolation source (fish, shrimp, others). 

In the “vegetables” category, isolates from “papaya” were the only ones that differed from those from other vegetable sources in their AMR profile, although it happened only for aminoglycoside, quinolone and fosfomycin resistance genes. However, considering the number of “pepper” isolates was big, we split this category into three groups:

4. Papaya, n=279

5. Pepper, n=216

6. Other vegetables, n=568

The “aquatic environments” category was split into two groups, based on differences between surface waters and water from irrigation canals and other sources: 

7. Surface water (rivers, ponds, lakes, dams), n=307

8. Other water sources (irrigation canals and other sources), n=246

Finally, two new categories were added:

9. Avian, n=193

10. Other sources, n=249

The methods section was updated to reflect these changes.

RC3. Authors should be able to report MLSTs of their isolates since they have already had WGS data. I believe this is highly important since AMR profiles and sub-serotypes are highly associated. 

AR3. Information on MLSTs is published in a previous paper on the epidemiology of Salmonella associated with bovine lymph nodes and ground beef [1]. This paper is cited in the introduction (L78-80). However, we agree to provide information on the STs of our experimental isolates in the subsection “Salmonella isolates” of the methodology, as well as to include STs in the discussion.

RC4.Another important aspect is the specific source of the 77 isolates should be provided along with the date of collection. These features need to be considered while driving any conclusion related to AMR profile and “source”. As a reviewer, I would like to see if the major group of Typhimurium isolates collected on 9/18/18 that show highly similar genotypic and phenotypic AMR profiles were all from the same batch of samples, city, or producer. Please clarify also in the manuscript and provide data.

AR4. In connection with the previous comment, we included detailed information on the origin of isolates. Isolates from this project originated from the same company that integrates feedlot and slaughterhouse operations in the state of Veracruz. They sell carcasses in a wholesale store in Mexico City. All the samples originated from crossbred Bos indicus young bulls 24-36 months old. All the animals were harvested in the same slaughterhouse, while samples were collected at the store. Thus, all isolates originated from the same place, not only those of Typhimurium serovar. But they were collected across a 2-year period and all the isolates originated from a different sample. We agreed to include this information in the methods and discussion sections, for the sake of clarity. Please, refer to the updated manuscript to see the changes.

RC5. Abstract: Authors did not talk about 3GC in the abstract. Please mention the function/importance of ramR gene for MDR. The authors did not discuss their findings for high levels of amoxicillin-clavulanic acid-resistant isolates in the absence of AMR genes related to this phenotype in the manuscript. 

AR5. Since resistance to 3GC and carbapenem was low, we did not mention them in the abstract in order to meet the maximum allowed length, while including the most relevant information. For the same reason, we did not expand on the role of ramR gene in MDR phenotypes, as well as other findings. Nevertheless, the abstract was updated to include 3GC and carbapenems results.

Regarding the role of ramR gene mutations on MDR phenotypes, we did mention it in the manuscript (L444-446) and cited the paper that has a thorough explanation of the topic. However, we agree to broaden the discussion on ramR mutations, including their role in resistance to betalactams in isolates lacking ESBLs, as pointed out by the reviewer

RC6. Lines 100-102 Authors should explain why they choose these individual lists of antibiotics while there are many other antibiotics/classes listed in WHO. AMR in Salmonella is especially important when resistance emerges for the antibiotics that are used to treat Salmonella infections in humans. Macrolides (e.g., azithromycin) are one of the important antibiotics to treat Salmonella infections and this antibiotic was not included in this study. How would the authors explain their selection of antibiotics is unclear in the manuscript? Please also provide the class information of selected antibiotics either in the manuscript or table/figures.

AR6. We included antibiotics that are in the WHO list of critically important and highly important antibiotics, according to WHO’s prioritization criteria [2]:

P1) High absolute number of people affected by diseases for which the antimicrobial is the sole or one of few therapies available.

P2) High frequency of use in human medicine.

P3) Evidence of transmission of resistant bacteria or AMR genes from non-human sources.

Thus, we picked some antimicrobials that are considered highest priority (i. e. aminoglycosides, carbapenems, quinolones), as well as those that are approved in Mexico for the treatment of both humans and food-producing animals (i. e. quinolones, 3GC, folate pathway inhibitors, etc.). Concerning azithromycin, it is only approved to treat cats and dogs in Mexico [3]. Moreover, according to a recent review covering the 2000-2017 period, azithromycin-resistant Salmonella has not been isolated from foods (including meats) in Mexico [4]. So, we decided to leave macrolides out of the AST panel. However, we know azithromycin resistant Salmonella is considered a public health risk in some countries, although it involves mainly typhoidal strains [5]. Hence, we limited macrolide analysis to comparative genomics, which confirmed there is a very low rate (<1%) of azithromycin resistant genotypes (mphAB and lnuF genes) among the public Salmonella isolates from Mexico studied here (n=2,405). This information was included in methods and results, while the discussion was broadened to include analysis of results on macrolide resistant genotypes. We also acknowledged this is a limitation of our study, considering the widespread distribution of acrB mutations, which have been reported to confer macrolide resistance in typhoidal strains [6] (refer to the “Discussion” section of the updated manuscript). Finally, the antibiotic class is not missing. We did provide that information in Fig. 1 label (L199-202 of the original manuscript). However, this figure was updated. We included all the details required for a proper interpretation of results, without having to consult the text.

RC7. Line 133- Please describe which parameter was used to detect the “poor quality reads used for trimming criteria

AR7. We used the FastQC program, as declared in L132. This program analyzes raw read files and reports the quality score (Q score) across all bases, among other attributes. The Q scores report the probability of error in base calling (P=10^((-Q)⁄10)). So, reads that are going to be used for genome assembly should have a Q score of 30, ideally. In that way, we have a 99.999% confidence that the base called in that position is right. Below we provide an example of this report:

As you see, there is drop in the quality of base calling by the end of the insert (around position 155 and onwards). This is typical of raw reads obtained with Illumina technology, which uses the “sequencing by synthesis” approach. This requires adding adaptors to DNA fragments. So, that they can attach to the flow cell for DNA amplification and cluster formation. So, the adapters shall be removed before genome assembly. At the beginning, the quality of base calling is very good, as there is plenty of reagents and thus, every cluster flashes simultaneously when a new base is added. However, as reagents are consumed, some DNA fragments in the same cluster flash before others (pre-phasing) and/or some others flash later (phasing). This causes a fall in the quality of base calling, since the equipment cannot easily identify to which base the flash corresponds. Hence, researchers should make sure low-quality reads are removed before conducting genome assembly.

We also want to clarify that the minimum quality criterium was mentioned in the manuscript (L134, Q≥30). We did not provide a thorough explanation since this is a standardized procedure that it is not normally explained in papers. Please, refer to a recent publication in Plos One where the authors do not even mention quality assessment of raw reads [7]. Instead, they just refer the use of Trimmomatic to remove low quality reads. So, we do not believe expanding on this topic is necessary.

RC8. Line 136- Please provide the version of the Spade tool. Did the authors conduct a quality check for their assemblies?

AR8. We included the Spades version in the methods section of the revised manuscript. The genome assembly service of PATRIC includes a QUAST quality report. This was also included in the methodology. We used to report that information in our papers, but it does not seem to be required by journals since anyone can download raw reads and reproduce the analysis. The Plos One paper we cited before [7] does not include this information either. However, we do have the relevant quality attributes of genome assembly (# of contigs, assembly length, GC, N50, L50) which were included in supplementary S1 file.

RC9. Line 147- Please refer to the human isolates as “human clinical cases” in related figures, tables, or manuscript as referred to in the line.

AR9. After adjusting the groups of isolates, we decided to name this group just as “human isolates”. This new name is easier to refer in the text and similar to other sources (i. e. avian, bovine, etc.).

RC10. Line-149- Please provide the search term used to find these isolates and the criteria used. So, the readers can reach the same data analyzed in this study.

AR10. Accepted.

RC11. Line 151- Please provide if the AMRFinderPlus database and program that was mentioned is the same one that is used in this analysis. Please provide the information of the data analyzed for this study that was scanned using the same database and version of the AMRFinderPlus. Since these results are prone to change as the AMR database and software are updated, authors must confirm that they compare the outcomes of WGS data using the same tools for all isolates included in the manuscript.

AR11. Noted. The NCBI Pathogen Detection site is connected to the National Database of Antibiotic Resistant Organisms (NDARO) and to the FDA’s Global Resistome Data-Resistome Tracker Tool (https://www.fda.gov/animal-veterinary/national-antimicrobial-resistance-monitoring-system/global-resistome-data). Hence, the results they publish on AMR genotypes are kept and updated by the NCBI team, as required. In spite of this, we checked to make sure changes to AMRFinderPlus program and databases did not impact results for AMR genes. The program has been improved to include prediction of point mutations, as well as biocide, stress, heavy metal resistance and virulence genes. Please, refer to the github AMRFinderPlus releases page (https://github.com/ncbi/amr/releases) for a thorough report of the changes made across versions. In addition, we tested whether results would change across versions. For instance, we got the same results for our 77 isolates by running AMRFinderPlus locally (version 3.8.4) as compared to those reported by NCBI for the same isolates (they used version 3.6.7). Likewise, we got the same results reported by NCBI for 32 isolates (analyzed with the earliest version 3.2.3 among isolates from Mexico) while running these genomes with version 3.8.4 locally. Taking this into consideration, we decided to add a brief explanation on AMR genotypes published by NCBI and the analysis we conducted to make sure AMR gene profiling is not affected by the program version or database, among this group of isolates.

RC12. Line 152- Please provide the methodology for serotyping

AR12. This information was already published in our previous research that we cited on the methods section [1]. We performed in silico analysis to predict serovar by running raw reads on the SeqSero software, version 1.2 [8]. However, for the sake of clarity, we added information on how serotyping was conducted in the methods section. Please, refer to the section “Salmonella isolates”.

RC13. Line 166- Please explain how the AMR profiles of each plasmid were determined.

AR13. Plasmid profiling is different from AMR profiling. As mentioned in the original manuscript (L167-169), we used PlasmidFinder program, version 2.1, to predict the plasmid profile of each isolate. If the program detected plasmid replicons in a genome, then we collected the reference sequence of each predicted plasmid from NCBI, which contains all the coding sequences of the plasmid, including those related to AMR. So, there is no need to determine the plasmid’s AMR profiles. Just by looking at their reference sequences we can determine if they harbor AMR genes. The objective of plasmid profiling is to check if the isolates are likely carrying plasmids. 

RC14. Line 171- Why the authors select the %70 thresholds?

AR14. The 70% cutoff is arbitrary. It aims to establish that the majority of a reference plasmid sequence must be present in a draft genome to propose that the genome actually carries that plasmid. WGS data includes both chromosomal and plasmid DNA (if the strain carries a plasmid). Although cumbersome, it is possible to predict the presence of plasmids in a draft genome by combining different lines of evidence:

1) Detection of plasmid replicons in the genomes 

2) Tiling contigs against reference plasmids. If we identified consecutive genes that were in the same genomic context in contigs and plasmids, and the majority of the reference plasmid is covered in the genome, these were proposed to be plasmids in that genome. 

3) Average depth of coverage. Normally, plasmid-associated contigs have the same or very similar read depths between them, and different from that of chromosomal contigs. There are few exceptions, such as multi-copy ribosomal genes, as well as big plasmids, which may have similar read depths as compared to chromosomal DNA. However, in general, this approach works fairy well and checking read depth is only required when contigs do not tile well against reference plasmids. We started to use this method based on a paper published before in Plos One [9]. It has been accepted in papers published by our research group in Scientific Reports [10] and Journal of Microbiology [11]. The manuscript provides accessions of both our draft genomes and plasmids. So, that anyone can reproduce the analysis. For the sake of clarity, we described the methodology in full and cited previous papers using the same approach.

RC15. Line 175-180 These lines are misleading. There is no phenotype data included in the analysis of publicly available data. Please modify it accordingly.

AR15. Accepted. The text was modified to make clear which analyses were conducted with our isolates and which ones with public isolates.

RC16. Line 181- Even though the statistical language is sometimes used loosely in this regard, the authors performed analyses of association - not correlation - between phenotype and genotype and they will probably wish to change the wording to reflect that important difference.

AR16. We calculated the Pearson correlation coefficient between the % of phenotypically non-susceptible isolates and the % of genotypically non-susceptible isolates for each of the tested antibiotics (Fig. 2). Based on the previous comment, we modified the wording of this whole section for the sake of clarity.

RC17. Line 512-513 I would suggest authors review their statement related to WGS is being a better tool to monitor aminoglycoside resistance as compared to AST. The area of aminoglycoside resistance is difficult and evolving and interpretations are difficult and changing. Nonetheless, there is apparent cryptic genes aac(6’) and other factors such as breakpoints set for resistance for which phenotype and genotype seem disconnected. This disagreement does not infer that WGS is better, assuming that in at least some cases the naming conventions for the genes themselves might be based on flawed original experimentation, or else based originally in another genus or species.

AR17. Accepted. That statement was removed from the text.

RC18. Line 520 Please provide how many isolates had ramR mutations and did not harbor AMR genes but were found phenotypically resistant to chloramphenicol to support this statement.

AR18. This statement is based on the strong association between chloramphenicol resistance and ramR mutations that we reported in the “Results” section of the original manuscript (L333-335). However, we agreed to provide detailed information on the predicted mutations for each isolate, as recommended by the reviewer (see the updated Fig 1).

RC19. Line 538-577 Authors may revise their justification related to the MDR presence in lymph node and ground beef as fecal origin Salmonella may travel from intestine to LN via payer’s patches, and transmission of environmental Salmonella may most likely occur via fly bites in the feedlots. Most importantly, lymph nodes are the most likely source of Salmonella found in ground beef products and contamination can occur via the fat trimming process.

AR19. We agree lymph nodes (LN) contribution to Salmonella contamination in ground beef is well documented. This is not questioned in the manuscript. However, we did find a strong association between source of isolates and the proportion of MDR strains (L215-217), with a higher probability of finding MDR isolates in ground beef. Moreover, LN are not the sole source of Salmonella in ground beef, especially in developing countries, where the rate of carcass contamination (which is mainly of fecal origin) is higher as compared to that observed in industrialized nations. For instance, there are Mexican studies reporting a Salmonella prevalence of 6-18% in beef carcasses [12-14], while in ground beef the rates may go above 60-70% [4, 13, 15]. Thus, fecal contamination is likely contributing to Salmonella contamination in ground beef in countries with a similar situation. This is further supported by our previous findings [1], showing some Salmonella serovars (i. e. Typhimurium) were only detected in ground beef samples but not in LN from the same carcass, while some others were exclusively present in LN (i. e. Kentucky, Fresno, Muenster, Give) and not in ground beef obtained from the same carcass. Nonetheless, we revised the wording of this section to temper the statements and make sure they are just used to put the research into context without implying a challenge to the well-known contribution of LN to ground beef Salmonella contamination.

RC20. Table 1 – Please explain “C”

AR20. There is a superscript in the footnote of Table 1 indicating it refers to the antimicrobial’s disk concentration.

RC21. Table 2- is confusing. Please consider reporting the point mutations and observed phenotype observed in 77 isolates that were not related to a defined AMR gene.

AR21. Accepted. This table was removed, and its data were incorporated in Fig 1, as suggested by the reviewer in other comments.

RC22. Table 3- how did the author screen the plasmidal content? Please include in the methodology. 

AR22. This information was included in the methodology (L166-173 of the original manuscript). We used PlasmidFinder program, version 2.1. Considering this comment, as well as RC14 above, we decided to expand the description of these methods in the updated manuscript.

RC23. Figure 1- This table is hard to read. Authors may consider placing the class of phenotypic and genotypic resistance data next to each other for each class. I believe the qacEDelta1 is irrelevant with this research and is not shown on the AMRFinderPlus outcome of the isolates based on my research for given SAM IDs. Please help me to understand how this gene was found and why it was related to this study. Please also add ramR column here since the relationship between the mutation and phenotype has been analyzed and discussed.

AR23. The table was updated considering this reviewer suggestions and we also included information on point mutations, as well as MDR phenotypes. Regarding the qacEDelta1 gene, it can be predicted from assembled genomes, together with other biocide, stress, and heavy metal resistance genes, if the analysis is run with the --plus option. Please, refer to the github AMRFinderPlus repository for usage options (https://github.com/ncbi/amr/wiki/Running-AMRFinderPlus#usage). In a previous study [11], we observed there was association between the presence of qacEDelta1 gene and MDR phenotypes. The stress induced by many factors, including biocides, is known to trigger AMR resistance in bacteria [16]. Moreover, some of these genes are frequently found in isolates carrying class-1 integrons, which are known to play a key role in bacterial AMR [37]. Thus, we aimed to assess if biocide resistance genes were disseminated among experimental isolates showing MDR phenotypes. So, we decided to run AMRFinderPlus with the --plus option, to check if there was association between the presence of this gene and the observed phenotypes. Nevertheless, considering this gene was present in just a few experimental and public isolates, we agree to remove it from the table.

RC24. Figure 2- Please provide the number of isolate information for each antibiotic. Even though the statistical language is sometimes used loosely in this regard, the authors performed analyses of association - not correlation - between phenotype and genotype and they will probably wish to change the wording to reflect that important difference.

Please consider using singular for sources. I would suggest revising the presentation of the source of isolates such as (e.g. human, bovine). Are these isolates have all genes listed in the conferring header (e.g., tetABCDGM) or at least one of them? Please clarify and add a footnote as needed.

AR24. The label of Figure 2 mentions the total number of isolates (n=77, L222 of the original manuscript) and the figure reports the % of isolates with phenotypic and genotypic resistance to each tested antibiotic. So, the number of isolates can be calculated. Moreover, we are reporting the results of a Pearson correlation analysis, as mention before. This was clarified in the methods section. To facilitate the interpretation of these results, however, we modified Figure 2 and it is now reporting the number of non-susceptible isolates in each antibiotic, instead of the proportion. Moreover, we included the Pearson correlation coefficient and the corresponding P-value in the graph. 

We understand the second part of this comment refers to Fig 3. Authors agreed to name sources in singular. Moreover, the source of every isolate was double checked to make sure the information is in accordance with the recorded isolation source at NCBI. For the heatmap, we grouped AMR alleles according to the resistance mechanism they encode. Regarding tetracycline resistance, we only detected AMR genes encoding efflux mechanisms (tetABCDGM). Thus, isolates were considered genotypically resistant if they carry at least one of these genes. We agree this needed clarification and thus, this figure was modified considering previous comments, as well as those stated here.

RC25. S1 Table- please revise the number of sample size in the tab. I would suggest removing the collection date from Table 1 since it was not discussed in the manuscript. Please also provide the date of isolation and if possible, the information about the source of isolates (if they were collected from the same city/abattoir etc.) of your isolates in S1.

AR25. The sample size in the tab is an editing mistake. Furthermore, in NCBI metadata, isolation date and collection date are the same. In fact, there is no “isolation date” field at the NCBI Pathogen Detection website. In connection with previous comments, the methods and discussion sections were modified to include information about the source of isolates. The influence of collection dates is discussed in our previous paper [1]. But it only refers to a seasonality in Salmonella prevalence, which is higher in warm climate as compared to winter. Therefore, we are now reporting the association between AMR profiles and season or year of collection. This information was not mentioned in the original manuscript. So, it was now incorporated in the results and discussion sections.

RC26. I would also suggest providing the phenotypic data in S1 along with the ramR mutation.

AR26. We do not accept this suggestion. Data on AMR phenotypes and mutations were incorporated in the main body of the manuscript. So, there is no need to duplicate that information. Thus, we decided to remove the AMR genotypes from S1 Table instead.

RC27. There are duplicated gene names (.e.g., sul1), please correct.

AR27. We would like to clarify that the duplicated sul1 gene in some isolates is not a mistake. This only happened in isolates of serovar Typhimurium carrying the resistance island SGI1. Please, refer to Figure 4 where you can check that this gene is actually duplicated in SGI1. This can also be checked by accessing the SGI1 reference sequence (AF261825.2), which is provided in Figure 4 as well.

However, AMR genotypes were removed from S1 Table to avoid duplications with figures from the main body of the manuscript. 

RC28. Please include the AMR related genes and mutations and isolate names as shown in Table 1 as they are not matching with supplemental material (the last column in Table 1 is missing.

AR28. We understand the reviewer refers to Figure 1 when citing Table 1. As mentioned before, AMR genotypes were removed from S1 Table to avoid duplicating information with the main body of the manuscript. The purpose of S1 Table is to comply with data availability requirements. All tables and figures were revised to make sure isolate names are used uniformly across the manuscript.

RC29. I would also suggest using the same isolate names consistent in the manuscript and related data.

AR29. Accepted. The tables were updated according to this comment.

RC30. Many cells in the spreadsheet are missing, please revise.

AR30. This table was prepared by downloading our own isolates from the NCBI Pathogen Detection website. While submitting raw reads to NCBI, some laboratories fail to indicate the serovar. In other cases, NCBI do not assemble some isolates. This is the reason why there are blank cells, corresponding to records with no serovar recorded in the metadata, as well as empty Assembly and WGS accessions, for those isolates that have not been assembled by NCBI so far. Again, since the purpose of S1 Table is to comply with data availability, we decided to provide the information as it appears at the public repository. However, we updated this table and left only relevant information that allows readers to reach the data.

RC31. There are grammatical issues that need to be corrected. Authors need to revise AMR gene names and make sure they are presented correctly and consistently in both the body manuscript and related data.

AR31. The manuscript was carefully revised to correct grammatical mistakes. We used gene names as reported by AMRFinderPlus and published at the NCBI pathogen detection website. Both the manuscript and supplementary files were checked to make sure they are correctly and consistently used.

RC32. S2 Table- Please correct the name on the tab. There are “environment or environmental sample” related isolates classified as the aquatic environment. Please revise to verify these isolates are actually from an aquatic source.

AR32. Noted. The name on the tab was left by mistake. It was removed. There are indeed some isolates recorded as “environment or environmental sample” that were collected from water samples, as indicated in their Biosample records. Nevertheless, according to previous comments on this issue, all source categories were revised, and corrections were made in the updated version of S2 Table.

RC33. S3 Table- Please revise the AMR genotype column as there are duplicates, and information not relevant to AMR genes

AR33. This table is reporting information as it appears at the NCBI Pathogen Detection website. When there are duplicated genes, this should be related to the presence of integrons containing duplicated AMR gene cassettes in that isolate. As occurs in the SGI1 integron commented before (see AR27), this is not a mistake.

RC34. Reference: Overall- References are not standardized, please correct the inconsistency observed with the lower- and upper-case use. Please revise all the links provided (e;g., URL of Ref 44 is not working) and provide accession dates for each URL. Please also italicize the spp names as needed.

AR34. References were generated automatically by using the last version of Plos reference style in EndNote X9.3.3, as recommended by Plos One. We also checked recent papers published at Plos One (i.e. DOI: https://doi.org/10.1371/journal.pone.0244057) and they also lack uniformity in the lower- and upper-case use. Probably, this inconsistency is caused by the way EndNote or similar programs capture the title of each publication. We thought this was acceptable considering there are published papers with these inconsistencies and some others. For instance, in the above referred paper, journal names are not abbreviated, which is supposed to be a Plos One requirement. Instead, they use a “Sentence case” style, with only the first word of the journal name capitalized and the text is italicized. In any case, we carefully reviewed this section to remove any mistake or inconsistency pointed by the reviewer. The URL of Ref 44 was recently updated by the government. We checked again every URL provided to make sure they all work and added the accession dates.

1. Palós Gutiérrez T, Rubio Lozano MS, Delgado Suárez EJ, Rosi Guzmán N, Soberanis Ramos O, Hernández Pérez CF, et al. Lymph nodes and ground beef as public health importance reservoirs of Salmonella spp. Revista Mexicana de Ciencias Pecuarias. 2020;11(3):795-810. doi: 10.22319/rmcp.v11i3.5516.

2. WHO. Critically important antimicrobials for human medicine. 6th Revision 2018: World Health Organization; 2019. Available from: https://www.who.int/foodsafety/publications/antimicrobials-sixth/en/.

3. SADER. Productos químico-farmacéuticos vigentes: Secretaría de Agricultura y Desarrollo Rural, Gobierno de México; 2020 [cited 2020 September 20, 2020]. Available from: https://www.gob.mx/cms/uploads/attachment/file/512374/PRODUCTOS_VIGENTES_QF_2019.pdf.

4. Godinez-Oviedo A, Tamplin ML, Bowman JP, Hernandez-Iturriaga M. Salmonella enterica in Mexico 2000-2017: Epidemiology, Antimicrobial Resistance, and Prevalence in Food. Foodborne Pathog Dis. 2020;17(2):98-118. Epub 2019/10/28. doi: 10.1089/fpd.2019.2627. PubMed PMID: 31647328.

5. McDermott PF, Zhao S, Tate H. Antimicrobial Resistance in Nontyphoidal Salmonella. Microbiol Spectr. 2018;6(4). Epub 2018/07/22. doi: 10.1128/microbiolspec.ARBA-0014-2017. PubMed PMID: 30027887.

6. Hooda Y, Sajib MSI, Rahman H, Luby SP, Bondy-Denomy J, Santosham M, et al. Molecular mechanism of azithromycin resistance among typhoidal Salmonella strains in Bangladesh identified through passive pediatric surveillance. PLoS Negl Trop Dis. 2019;13(11):e0007868. Epub 2019/11/16. doi: 10.1371/journal.pntd.0007868. PubMed PMID: 31730615; PubMed Central PMCID: PMCPMC6881056.

7. van den Berg RR, Dissel S, Rapallini MLBA, van der Weijden CC, Wit B, Heymans R. Characterization and whole genome sequencing of closely related multidrug-resistant Salmonella enterica serovar Heidelberg isolates from imported poultry meat in the Netherlands. PLOS ONE. 2019;14(7):e0219795. doi: 10.1371/journal.pone.0219795.

8. Zhang S, Yin Y, Jones MB, Zhang Z, Deatherage Kaiser BL, Dinsmore BA, et al. Salmonella serotype determination utilizing high-throughput genome sequencing data. J Clin Microbiol. 2015;53(5):1685-92. doi: 10.1128/JCM.00323-15. PubMed PMID: 25762776; PubMed Central PMCID: PMCPMC4400759.

9. Dhanani AS, Block G, Dewar K, Forgetta V, Topp E, Beiko RG, et al. Genomic Comparison of Non-Typhoidal Salmonella enterica Serovars Typhimurium, Enteritidis, Heidelberg, Hadar and Kentucky Isolates from Broiler Chickens. PLoS One. 2015;10(6):e0128773. doi: 10.1371/journal.pone.0128773. PubMed PMID: 26083489; PubMed Central PMCID: PMCPMC4470630.

10. Delgado-Suárez EJ, Selem-Mojica N, Ortiz-López R, Gebreyes WA, Allard MW, Barona-Gómez F, et al. Whole genome sequencing reveals widespread distribution of typhoidal toxin genes and VirB/D4 plasmids in bovine-associated nontyphoidal Salmonella. Sci Rep. 2018;8(1):9864. doi: 10.1038/s41598-018-28169-4.

11. Delgado Suárez EJ, Ortíz López R, Gebreyes WA, Allard MW, Barona-Gomez F, Salud Rubio MS. Genomic surveillance links livestock production with the emergence and spread of multi-drug resistant non-typhoidal Salmonella in Mexico. J Microbiol. 2019;57(4). doi: DOI 10.1007/s12275-019-8421-3.

12. Narvaez-Bravo C, Miller MF, Jackson T, Jackson S, Rodas-Gonzalez A, Pond K, et al. Salmonella and Escherichia coli O157:H7 Prevalence in Cattle and on Carcasses in a Vertically Integrated Feedlot and Harvest Plant in Mexico. J Food Prot. 2013;76(5):786-95.

13. Martinez-Chavez L, Cabrera-Diaz E, Perez-Montano JA, Garay-Martinez LE, Varela-Hernandez JJ, Castillo A, et al. Quantitative distribution of Salmonella spp. and Escherichia coli on beef carcasses and raw beef at retail establishments. Int J Food Microbiol. 2015;210:149-55. Epub 2015/07/01. doi: 10.1016/j.ijfoodmicro.2015.06.016. PubMed PMID: 26125489.

14. Perez-Montaño JA, González-Aguilar D, Barba J, Pacheco-Gallardo C, Campos-Bravo CA, García S, et al. Frequency and Antimicrobial Resistance of Salmonella Serotypes on Beef Carcasses at Small Abattoirs in Jalisco State, Mexico. J Food Prot. 2012;75(5):867-73. doi: 10.4315/0362-028X.JFP-11-423.

15. Cabrera-Diaz E, Barbosa-Cardenas CM, Perez-Montano JA, Gonzalez-Aguilar D, Pacheco-Gallardo C, Barba J. Occurrence, serotype diversity, and antimicrobial resistance of salmonella in ground beef at retail stores in Jalisco state, Mexico. J Food Prot. 2013;76(12):2004-10. doi: 10.4315/0362-028X.JFP-13-109. PubMed PMID: 24290673.

16. Poole K. Bacterial stress responses as determinants of antimicrobial resistance. J Antimicrob Chemother. 2012;67(9):2069-89. doi: 10.1093/jac/dks196. PubMed PMID: 22618862.

---

## [Decision Letter · Decision Letter 1]

19 Mar 2021

PONE-D-20-36934R1

Genomic surveillance of antimicrobial resistance shows cattle are a moderate source of multi-drug resistant non-typhoidal Salmonella in Mexico

PLOS ONE

Dear Dr. Delgado-Suárez,

Thank you for submitting your manuscript to PLOS ONE. After careful consideration, we feel that it has merit but does not fully meet PLOS ONE’s publication criteria as it currently stands. Therefore, we invite you to submit a revised version of the manuscript that addresses the points raised during the review process.

There are still some mismatches between data and statements in the manuscript. Also there are missing points in the supplementary Table. Please address all points raised by the reviewers.

We look forward to receiving your revised manuscript.

Kind regards,

Iddya Karunasagar

Academic Editor

PLOS ONE

Journal Requirements:

Additional Editor Comments (if provided):

The reviewers have pointed out some mismatches between the statements and data including those in supplementary Table. Please correct these discrepancies and other points raised by the reviewers.

Reviewers' comments:

Reviewer's Responses to Questions

**Comments to the Author**

1. If the authors have adequately addressed your comments raised in a previous round of review and you feel that this manuscript is now acceptable for publication, you may indicate that here to bypass the “Comments to the Author” section, enter your conflict of interest statement in the “Confidential to Editor” section, and submit your "Accept" recommendation.

Reviewer #1: All comments have been addressed

2. Is the manuscript technically sound, and do the data support the conclusions?

Reviewer #1: Yes

3. Has the statistical analysis been performed appropriately and rigorously? 

Reviewer #1: Yes

4. Have the authors made all data underlying the findings in their manuscript fully available?

Reviewer #1: Yes

5. Is the manuscript presented in an intelligible fashion and written in standard English?

Reviewer #1: Yes

6. Review Comments to the Author

Reviewer #1: Fig 1. Some of the phenotypic resistance (e.g., STX) related cells were highlighted (filled), please be consistent. This is not matching with author’s statement for Fig 1. “ For AMR genotypes, cells filled with the corresponding antibiotic class color indicate the gene is present.” Authors may also consider rearranging the order of the isolates by serotype followed by day and sample type. So, the readers can associate the time of collection and sample type data along with the resistance profiles observed.

Please correct “metadata” in “S3 File. NCBI accesions, metada and antimicrobial resistance genotypes of fully sequenced public Salmonella enterica ser. Typhimurium isolates from Mexico included in this study”. Also, please check other instances.

There are few S. Typhi included in the supplemental data, since the manuscript is about NTS, please revise accordingly.

S2 file. There are about 112 isolates that have missing AMR data, please either remove or justify adding these strains in the metadata.

In the s2 file, there are 46 Typhimurium isolates from Mexico, authors include 38 of them in the S3 file. Did I miss anything? I am asking this question based on their statement: “Furthermore, considering the epidemiological importance of this serovar, we also analyzed the whole set of Typhimurium isolates from Mexico deposited at NCBI (n=38, refer to S3 Table for accession numbers and AMR genotypes of this group of isolates).”

I am also having difficulty matching the selected genomes (n=77) from the previous study (ref #18), with the current study. Authors state: “In the present investigation, we conducted antibiotic susceptibility testing and WGS of 77 NTS isolates collected in the course of a previous research project involving bovine lymph nodes (n=800) and ground beef (n=745) across a two-year sampling period [18]” and also states “ We identified nine Salmonella serovars: Anatum (n=23), Reading (n=22), Fresno (n=4), Typhimurium (n=10), London (n=9), Kentucky (n=6), and Muenster, Give and monophasic Typhimurium 1,4,[5],12:i:- (one each)”. However, the previous paper referred by authors states “78 isolates obtained from the 1,545 samples analyzed in the two years” and in the same paper, there are Reading(n=23), Anatum (n= 23), Typhimurium (n= 11), London (n= 9), Muenster (n= 2), Kentucky (n= 5), Give (n=1), and Fresno (n=4) serotypes. There was also no monophasic serovar identified in the previous manuscript. In this current version – if authors claim they include isolates from previous study – they should clarify why there is a mismatch exist with the isolates corresponding. I see in the S1 file, the strain UNAM2018123_Sa_AN13 was marked as monophasic, however, in the S2 file, this strain was recorded as Typhimurium. And finally, at the S3 file, while all Typhimurium isolates were included from “Mexico” based on the metadata (S2 file), this was excluded. I am really confused. If this strain was later identified (or corrected) as monophasic, how the authors make sure the Typhimurium strains in the S3 file are all monophasic? Please clarify this for me if there is a misunderstanding or if I am missing anything, otherwise, please provide your selection criteria and your confirmation method used for strains reported in the S3 file.

Once again, please fill the blank cells in the metadata as either “not reported” or provide a footnote for those cells in all S-related files. It is important for readers to understand why the cells were left blank.

Authors stated “ For instance, most serovar Typhimurium isolates (9/10) carried SGI1”. However, in Fig 4, all 10/10 isolates were showing SGI1. Please clarify this.

Please consider to include the SGI1, AMR and phenotype info along with the ramR mutations in S1 file, it is very hard for the readers to match individual IDs and serotypes with corresponding data in the S1.

S1.File genome size cells need correction.

Please also revise the abbreviations used and use the original names at the first instances.

7. PLOS authors have the option to publish the peer review history of their article (what does this mean?). If published, this will include your full peer review and any attached files.

Reviewer #1: No

---

## [Author Response · Author response to Decision Letter 1]

24 Mar 2021

PONE-D-20-36934R1

Genomic surveillance of antimicrobial resistance shows cattle are a moderate source of multi-drug resistant non-typhoidal Salmonella in Mexico

PLOS ONE 

The authors thank the academic editor and the reviewer for their detailed review. Below we list each comment raised during the reviewing process, followed by the authors’ responses.

Academic Editor Comments (AEC)

AEC1. Journal Requirements

Authors’ Response (AR)1. As far as we know, we did not cite retracted papers. However, we carefully reviewed each cited reference again and none have been retracted. We found five papers that have been corrected but just for editing mistakes that do not affect the citation made in our manuscript. We also reviewed the reference list to make sure it is complete and correct. Please, refer to the revised manuscript. Finally, we added two new references (#24 and 25 in the revised version):

24. Zhang S, den Bakker HC, Shaoting L, Chen J, Dinsmore BA, Lane C, et al. SeqSero2: rapid and improved Salmonella serotype determination using whole genome sequencing data. Appl Environ Microbiol. 2019;85:e01746-19. doi: https://doi.org/10.1128/AEM.01746-19. PMID: 31540993

25. Yoshida CE, Kruczkiewicz P, Laing CR, Lingohr EJ, Gannon VPJ, Nash JHE, et al. The Salmonella in silico typing resource (SISTR): an open web-accessible tool for rapidly typing and subtyping draft Salmonella genome assemblies. PLoS One. 2016;11(1):e0147101. doi: 10.1371/journal.pone.0147101. PMID: 26800248

These references were added in connection with RC6 and AR6. 

AEC2. The reviewers have pointed out some mismatches between the statements and data including those in supplementary Table. Please correct these discrepancies and other points raised by the reviewers.

AR2. The authors appreciate the time invested by the reviewers to make a thorough revision of our manuscript. Comments will be taken into consideration to improve the paper and will be responded point by point.

Reviewer Comments (RC)

RC1. Fig 1. Some of the phenotypic resistance (e.g., STX) related cells were highlighted (filled), please be consistent. This is not matching with author’s statement for Fig 1. “ For AMR genotypes, cells filled with the corresponding antibiotic class color indicate the gene is present.” Authors may also consider rearranging the order of the isolates by serotype followed by day and sample type. So, the readers can associate the time of collection and sample type data along with the resistance profiles observed.

AR1. Accepted. These editing mistakes were corrected in the revised version. Regarding the rearrangement of isolates, they were already ordered by serovar and collection date. It is not possible to include sample type as another ordering criterium (for the whole set) without changing the previous order by collection date. Hence, the rearrangement was conducted within each serovar and now the collection date is ordered within sample type.

RC2. Please correct “metadata” in “S3 File. NCBI accesions, metada and antimicrobial resistance genotypes of fully sequenced public Salmonella enterica ser. Typhimurium isolates from Mexico included in this study”. Also, please check other instances.

AR2. Accepted. We corrected this editing mistake in S3 File and also reviewed all the supplementary files to make sure there were no overseen editing mistakes left.

RC3. There are few S. Typhi included in the supplemental data, since the manuscript is about NTS, please revise accordingly.

AR3. We acknowledge typhoidal strains, as human-restricted pathogens, are not commonly transmitted through food. However, in developing countries (such as Mexico), there is a high number of street vendors with poor hygienic practices and practically no health control. Hence, it is not possible to discard that some typhoidal strains involved in clinical cases may be foodborne. Especially, considering the Mexican surveillance system does not have attribution data. This is the reason why we included typhoidal strains in the analysis. However, removing these strains from the analysis will not change results, while improving the clarity of the manuscript in regard to its scope. Hence, we accepted to remove typhoidal strains from the analysis and Fig 3 was updated accordingly.

RC4. S2 file. There are about 112 isolates that have missing AMR data, please either remove or justify adding these strains in the metadata.

AR4. This is not a mistake. These isolates were not predicted to carry any AMR genes. That is why their AMR genotype cells were empty. As mentioned in the previous review round, this report matches the structure of metadata reported at the NCBI pathogen detection site. For the sake of clarity, we added the following note at the heading of this file: “Metadata is reported as recorded at the NCBI pathogen detection website. Blank cells in any column indicate this information is not available at NCBI. Isolates with blank cells under the AMR genotypes column were not predicted to carry any AMR gene”.

RC5. In the S2 file, there are 46 Typhimurium isolates from Mexico, authors include 38 of them in the S3 file. Did I miss anything? I am asking this question based on their statement: “Furthermore, considering the epidemiological importance of this serovar, we also analyzed the whole set of Typhimurium isolates from Mexico deposited at NCBI (n=38, refer to S3 Table for accession numbers and AMR genotypes of this group of isolates).”

AR5. While recovering Typhimurium isolates from Mexico at the NCBI pathogen detection site, we used the plain serovar name “Typhimurium”. We did not include any variant, such as var. Copenhagen (there are four isolates of this variant in S2 File) or the monophasic 1,4,[5],12:i:-. One of our isolates is monophasic but is recorded as Typhimurium at NCBI. At the moment this isolate was submitted to NCBI by the Mexican Department of Agriculture, it was predicted as Typhimurium, based on in silico analysis of raw reads. As we will explain later in this document (see AR6), after assembling genomes and running serovar prediction analyses with SeqSero2 and SISTR programs, the serovar of this isolate was corrected to monophasic 1,4,[5],12:i:-. Hence, there are five isolates in S2 file that are Typhimurium variants and thus, they were not included in the Typhimurium set. 

Following this comment, we carefully checked S2 File and there are 45 records (not 46) that contain Typhimurium in the serovar field. After removing the variants, there are 40 Typhimurium instead of 38. During the previous review round, we updated the whole dataset. As a result, the number of isolates identified increased in almost all categories. Particularly, there were two additional Typhimurium isolates of human origin that were added and we forgot to update S3 File accordingly. This is the reason why we have the mismatching. We thank you for the detailed revision. The methods section was updated to indicate we identified 40 Typhimurium isolates (instead of 38). Likewise, the results section and S3 file were adjusted to reflect these changes and eliminate the mismatching.

RC6. I am also having difficulty matching the selected genomes (n=77) from the previous study (ref #18), with the current study. Authors state: “In the present investigation, we conducted antibiotic susceptibility testing and WGS of 77 NTS isolates collected in the course of a previous research project involving bovine lymph nodes (n=800) and ground beef (n=745) across a two-year sampling period [18]” and also states “ We identified nine Salmonella serovars: Anatum (n=23), Reading (n=22), Fresno (n=4), Typhimurium (n=10), London (n=9), Kentucky (n=6), and Muenster, Give and monophasic Typhimurium 1,4,[5],12:i:- (one each)”. However, the previous paper referred by authors states “78 isolates obtained from the 1,545 samples analyzed in the two years” and in the same paper, there are Reading(n=23), Anatum (n= 23), Typhimurium (n= 11), London (n= 9), Muenster (n= 2), Kentucky (n= 5), Give (n=1), and Fresno (n=4) serotypes. There was also no monophasic serovar identified in the previous manuscript. In this current version – if authors claim they include isolates from previous study – they should clarify why there is a mismatch exist with the isolates corresponding.

AR6. Our previous study (ref #18) was written and published when we only had the raw reads available. While using raw reads in SeqSero for serovar prediction, the monophasic Typhimurium isolate was predicted as Typhimurium. After assembling genomes, we repeated the analysis for all isolates both with SeqSero2 and SISTR programs (two references #24 and 25 were included to cite the source of both. programs). By doing so, we confirmed serovar prediction results, which were mostly consistent with results reported in ref#18, except for the monophasic Typhimurium and one isolate predicted as serovar Reading that we had to discard for having poor assembly quality and inconsistent results with Salmonella species (genome size >8 Mb and GC content of 46.5%). Due to the COVID-19 pandemia, this isolate has not been re-sequenced yet. We are currently waiting for re-sequencing results before filing an amendment request to the publisher of ref#18. In the meantime, however, we decided to exclude this isolate from the present manuscript. This clarification was included in the methods section.

RC7. I see in the S1 file, the strain UNAM2018123_Sa_AN13 was marked as monophasic, however, in the S2 file, this strain was recorded as Typhimurium. And finally, at the S3 file, while all Typhimurium isolates were included from “Mexico” based on the metadata (S2 file), this was excluded. I am really confused. If this strain was later identified (or corrected) as monophasic, how the authors make sure the Typhimurium strains in the S3 file are all monophasic? Please clarify this for me if there is a misunderstanding or if I am missing anything, otherwise, please provide your selection criteria and your confirmation method used for strains reported in the S3 file.

AR7. We are sorry to have caused such a confusion. S2 file contains information as recorded at the NCBI pathogen detection site. When isolates included in this manuscript were uploaded to NCBI, the submitter (the Mexican Department of Agriculture) included the serovar results we had at that moment. Therefore, isolate UNAM2018123-Sa-AN13 was recorded as Typhimurium. However, while preparing the current manuscript, and redoing serovar prediction analyses with assembled genomes in two different programs, we confirmed this isolate is indeed monophasic Typhimurium. That is why S1 File reports the correct serovar for this isolate. S3 File only reports Typhimurium isolates (not variants) available at NCBI. We did not change this record in S2 file for it would not match the actual record at NCBI. However, we already asked the Mexican Department of Agriculture to file an amendment request to NCBI, so that this record is corrected. Therefore, we decided to correct this record in S2 File as well, to avoid confusing the readers. In the updated version of the manuscript, including supplementary information, isolate UNAM2018123-Sa-AN13 is reported as monophasic Typhimurium. Moreover, we added the following note at the heading of the bovine isolates section of S2 File: “Isolate with Biosample accession SAMN12857424 (strain UNAM2018123_Sa_AN13) was recorded at NCBI as serovar Typhimurium at the moment its raw reads were submitted. However, further analyses with the assembled genome showed it is a monophasic Typhimurium variant (1,4,[5],12:i:-). This record should be corrected soon at the NCBI pathogen detection site.”

RC8. Once again, please fill the blank cells in the metadata as either “not reported” or provide a footnote for those cells in all S-related files. It is important for readers to understand why the cells were left blank.

AR8. As mentioned in AR4, we included a note below the heading of S2 file. We have also included a similar note below the heading of S3 files: “Metadata is reported as recorded at the NCBI pathogen detection website. Blank cells in any column indicate this information is not available at NCBI”. S1 file does not have any blank spaces.

RC9. Authors stated “ For instance, most serovar Typhimurium isolates (9/10) carried SGI1”. However, in Fig 4, all 10/10 isolates were showing SGI1. Please clarify this.

AR9. According to Fig 4, only 9 isolates carried SGI1. Please, re-check. Fig 4 

includes the 10 Typhimurium isolates as well as the monophasic variant. We included the variant in this analysis since it resisted several antimicrobials and we wanted to assess if it also carried SGI1. Please, notice there is one Typhimurium isolate (AK68) and the monophasic variant (AN13) lacking SGI1 (first two rings after the backbone). After these two, there are only 9 rings, corresponding to the remaining Typhimurium isolates.

RC10. Please consider to include the SGI1, AMR and phenotype info along with the ramR mutations in S1 file, it is very hard for the readers to match individual IDs and serotypes with corresponding data in the S1.

AR10. We do not see the need to duplicate the information in the main body of the manuscript and supporting files. In the main body, Fig 1 reports AMR phenotypes and mutations, as suggested by the reviewer previously. Likewise, Fig 4 reports the isolates carrying SGI1. Both figures report isolate names and serovars, which are also included in S1 File. Hence, we do not see why it would be difficult for readers to match individual IDs and serovars with information reported in S1 file. As mentioned in the previous review round, S1 File is meant to comply with data availability and facilitate reproducibility.

RC11. S1.File genome size cells need correction.

AR11. Our Excel program uses Spanish as the default number format (apostrophes to separate millions and commas to separate thousands). We corrected the file to comply with US format.

RC12. Please also revise the abbreviations used and use the original names at the first instances.

AR12. Accepted. We carefully reviewed the manuscript to make sure original names are provided at the first instances. In some cases, such as in figures and tables headings, original names and abbreviations are repeated to facilitate the interpretation of figures and tables without having to consult the text.

---

## [Decision Letter · Decision Letter 2]

9 Apr 2021

PONE-D-20-36934R2

Genomic surveillance of antimicrobial resistance shows cattle are a moderate source of multi-drug resistant non-typhoidal Salmonella in Mexico

PLOS ONE

Dear Dr. Delgado-Suárez,

Thank you for submitting your manuscript to PLOS ONE. After careful consideration, we feel that it has merit but does not fully meet PLOS ONE’s publication criteria as it currently stands. Therefore, we invite you to submit a revised version of the manuscript that addresses the points raised during the review process.

Please modify title by adding "poultry" 

We look forward to receiving your revised manuscript.

Kind regards,

Iddya Karunasagar

Academic Editor

PLOS ONE

Journal Requirements:

Additional Editor Comments (if provided):

Please modify title to include poultry as recommended by the reviewer.

Reviewers' comments:

Reviewer's Responses to Questions

**Comments to the Author**

1. If the authors have adequately addressed your comments raised in a previous round of review and you feel that this manuscript is now acceptable for publication, you may indicate that here to bypass the “Comments to the Author” section, enter your conflict of interest statement in the “Confidential to Editor” section, and submit your "Accept" recommendation.

Reviewer #1: All comments have been addressed

2. Is the manuscript technically sound, and do the data support the conclusions?

Reviewer #1: Yes

3. Has the statistical analysis been performed appropriately and rigorously? 

Reviewer #1: Yes

4. Have the authors made all data underlying the findings in their manuscript fully available?

Reviewer #1: Yes

5. Is the manuscript presented in an intelligible fashion and written in standard English?

Reviewer #1: Yes

6. Review Comments to the Author

Reviewer #1: (No Response)

7. PLOS authors have the option to publish the peer review history of their article (what does this mean?). If published, this will include your full peer review and any attached files.

Reviewer #1: No

---

## [Author Response · Author response to Decision Letter 2]

12 Apr 2021

PONE-D-20-36934R2

Genomic surveillance of antimicrobial resistance shows cattle and poultry are a moderate source of multi-drug resistant non-typhoidal Salmonella in Mexico

PLOS ONE 

The authors thank the academic editor and the reviewer for their detailed review. Below we list each comment raised during the reviewing process, followed by the authors’ responses.

Reviewer Comments (RC)

RC1. I believe the title of the manuscript needs a small revision (beside cattle, addressing the poultry as well as one of the sources based on their edits on data and results). At the discussion, the authors also acknowledge this by stating: " Likewise, our comparative analysis showed NTS strains isolated from cattle and poultry have strong AMR genotypes, which are similar to that of human clinical isolates.". I think the manuscript is almost ready for publication..

AR1. Accepted. The title was modified to include poultry. Please, refer to the revised version of the manuscript.

---

## [Editor Report · Decision Letter 3]

15 Apr 2021

Genomic surveillance of antimicrobial resistance shows cattle and poultry are a moderate source of multi-drug resistant non-typhoidal Salmonella in Mexico

PONE-D-20-36934R3

Dear Dr. Delgado-Suárez,

We’re pleased to inform you that your manuscript has been judged scientifically suitable for publication and will be formally accepted for publication once it meets all outstanding technical requirements.

Kind regards,

Iddya Karunasagar

Academic Editor

PLOS ONE

Additional Editor Comments (optional):

All reviewer comments have been addressed.
---

## [Editor Report · Acceptance letter]

21 Apr 2021

PONE-D-20-36934R3 

Genomic surveillance of antimicrobial resistance shows cattle and poultry are a moderate source of multi-drug resistant non-typhoidal *Salmonella* in Mexico 

Dear Dr. Delgado-Suárez:

I'm pleased to inform you that your manuscript has been deemed suitable for publication in PLOS ONE. Congratulations! Your manuscript is now with our production department. 

Kind regards, 

on behalf of

Dr. Iddya Karunasagar 

Academic Editor

PLOS ONE